# TRANSFER AND MARGINALIZE: EXPLAINING AWAY LABEL NOISE WITH PRIVILEGED INFORMATION

## ABSTRACT

Supervised learning datasets often have privileged information, in the form of features which are available at training time but are not available at test time e.g. the ID of the annotator that provided the label. We argue that privileged information is useful for explaining away label noise, thereby reducing the harmful impact of noisy labels. We develop a simple and efficient method for supervised neural networks: it transfers via weight sharing the knowledge learned with privileged information and approximately marginalizes over privileged information at test time. Our method, **TRAM** (TRansfer and Marginalize), has minimal training time overhead and has the same test time cost as not using privileged information. TRAM performs strongly on CIFAR-10H, ImageNet and Civil Comments benchmarks.

## 1 INTRODUCTION

Supervised learning problems are typically formalized as learning a conditional distribution $p(y|\boldsymbol{x})$, $y \in \mathcal{Y}$ and $\boldsymbol{x} \in \mathcal{X}$ from $(\boldsymbol{x}_i, y_i)$, $i = 1, ..., N$ pairs. Yet we often have access to additional features $\boldsymbol{a} \in \mathcal{A}$ at training time that will not be available at test time. These features are known as *privileged information* (Vapnik & Vashist, 2009), or **PI** for short. An example of PI are features describing the human annotator that provided a given label, such as the annotator ID, the length of time to provide the label, the experience of the annotator, etc. Annotators do not always agree on the correct label for a given $\boldsymbol{x}$, some annotators may be more reliable than others and the reliability of annotators may depend on the location of $\boldsymbol{x}$ in the input domain $\mathcal{X}$ (Snow et al., 2008; Sheng et al., 2008).

The expanded training dataset consists of $(\boldsymbol{x}_i, \boldsymbol{a}_i, y_i)$ triplets. Given that our test time predictive distribution cannot be conditioned on $\boldsymbol{a}$, what use is this PI? As a thought experiment, suppose there exists a malicious (or lazy) annotator that provides random labels. It is known that random labels harm the performance of supervised learning models (Frénay & Verleysen, 2013; Song et al., 2020; Cordeiro & Carneiro, 2020). If these random labels can be explained away via access to PI, such as the annotator ID, then this harm can be prevented. In particular we can use the PI to explain away noise in the labels which otherwise would be irreducible aleatoric uncertainty.

More formally, suppose $\boldsymbol{A}$, the PI random variable, is predictive of $\boldsymbol{Y}$ given $\boldsymbol{X}$, in the sense that the conditional mutual information $I(\boldsymbol{Y}; \boldsymbol{A}|\boldsymbol{X})$ is non-zero. Then, the entropy of $\boldsymbol{Y}$ is reduced if we condition on *both* $\boldsymbol{X}$ and $\boldsymbol{A}$ rather than $\boldsymbol{X}$ alone, as summarised in Lemma 1.1.

**Lemma 1.1.** $I(\boldsymbol{Y}; \boldsymbol{A}|\boldsymbol{X}) > 0 \Rightarrow H(\boldsymbol{Y}|\boldsymbol{X}, \boldsymbol{A}) < H(\boldsymbol{Y}|\boldsymbol{X})$.

In §2.1 we examine this lemma for a particular model, proving that under certain conditions, PI can be leveraged to lower the expected risk for linear regression problems. Additionally, prior work has proven that PI can lead to generalization bounds with better sample complexity (Vapnik & Vashist, 2009; Lambert et al., 2018).

Inspired by prior work and our theoretical analysis of a simple linear model, we focus on exploiting PI in supervised deep neural networks. The production deployment of such models often has tight latency and memory constraints. Hence a number of methods have been developed to utilize PI with the same test time memory and computation cost as networks trained without PI (Yang et al., 2017; Lambert et al., 2018; Lopez-Paz et al., 2015). Yang et al. (2017) uses PI as a form of input-dependent regularizer. Lambert et al. (2018) train with heteroscedastic Gaussian dropout, with the training-time dropout variance a function of the PI. Lopez-Paz et al. (2015) distill a network trained with PI into a network without access to $\boldsymbol{a}$.

Below we develop a method, TRAM, which transfers knowledge via weight sharing from the part of the network trained using PI to the test time network which does not have access to PI. At test time, TRAM makes a simple, efficient approximation to the integral $p(y|\boldsymbol{x}) = \int p(y|\boldsymbol{x}, \boldsymbol{a})p(\boldsymbol{a}|\boldsymbol{x})d\boldsymbol{a}$. Making predictions without PI is no more costly than that with a standard network trained without access to PI. Unlike prior work which requires specific techniques such as Gaussian Dropout, we need not constrain the form of the predictors to make the downstream marginalization possible. Implementation and training are simple.

In summary the paper contributions are the following:

- To better illustrate when PI is useful, we show analytically that, under certain conditions, PI reduces the expected risk for linear regression models.
- We provide empirical evidence suggesting that the representations learned with access to PI are more robust against label noise.
- We propose a novel efficient method, TRAM, which exploits PI in supervised deep neural networks and has zero computational overhead at prediction time.
- Empirically, we show that our method performs better than a series of baselines on CIFAR-10H, ImageNet and CivilComments benchmarks.

## 2 Exploiting Privileged Information

To build up intuition and to better illustrate situations where PI can be useful, we start with a simple linear model where a formal analysis can be carried out. Next, we look into non-linear models and provide a motivating experiment suggesting that useful PI can be leveraged in deep networks to improve representation learning.

### 2.1 When can PI be helpful? An analysis in a Simple Linear Model

We consider the following regression generative model with target $y$

$$y = \boldsymbol{x}^\top \boldsymbol{w}^\star + \boldsymbol{a}^\top \boldsymbol{v}^\star + \varepsilon$$

where $\boldsymbol{x} \in \mathbb{R}^d$ and $\boldsymbol{a} \in \mathbb{R}^m$ correspond to standard and PI features respectively, while $\varepsilon \sim \mathcal{N}(0, \sigma^2)$ stands for some additive noise. The two unknown parameters $(\boldsymbol{w}^\star, \boldsymbol{v}^\star)$ establish the relationships between the target $y$ and the features $(\boldsymbol{x}, \boldsymbol{a})$. To model the fact that the PI features can themselves depend on the $\boldsymbol{x}$—e.g., raters having diverging assessments on ambiguous input samples—we assume that $\boldsymbol{a} \sim p(\boldsymbol{a}|\boldsymbol{x}) = \mathcal{N}(\mu(\boldsymbol{x})|\Sigma(\boldsymbol{x}))$ for some mean and covariance *dependent on $\boldsymbol{x}$*.

Let us assume we have $n$ observations from this generative model represented by $\boldsymbol{y} \in \mathbb{R}^n$, $\boldsymbol{X} \in \mathbb{R}^{n \times d}$, $\boldsymbol{A} \in \mathbb{R}^{n \times m}$ and $\mu(\boldsymbol{X}) \in \mathbb{R}^{n \times m}$. We are interested in comparing different predictors $\tau(\boldsymbol{X})$ that can predict $y$ *only* based on $\boldsymbol{X}$, as required in the case of PI. To compare the predictors, we use the concept of risk $\mathbb{E}_{\varepsilon \sim p(\varepsilon), \boldsymbol{a} \sim p(\boldsymbol{a}|\boldsymbol{x})}[\mathcal{R}(\tau(\boldsymbol{X}))]$, formally defined in Appendix E, to capture the expected error of $\tau$ in predicting $y$; see Section 3.5 in Bach (2021) for more background about risk analysis.

We defer to Appendix E a rigorous exposition of the results and convey instead here some intuitive messages. We first focus on the comparison between

- (NO-PI) the least-square estimate $\hat{\boldsymbol{w}}_0 = (\boldsymbol{X}^\top \boldsymbol{X})^{-1}\boldsymbol{X}^\top \boldsymbol{y}$ that ignores $\boldsymbol{A}$, and
- (PI) the joint least-square estimate $[\hat{\boldsymbol{w}}_1; \hat{\boldsymbol{v}}_1] = (\boldsymbol{Q}^\top \boldsymbol{Q})^{-1}\boldsymbol{Q}^\top \boldsymbol{y}$ with $\boldsymbol{Q} = [\boldsymbol{X}, \boldsymbol{A}]$ in $\mathbb{R}^{n \times (d+m)}$. At prediction time, if we had access to $(\boldsymbol{x}_{\text{test}}, \boldsymbol{a}_{\text{test}})$ for (PI), we would predict with $\hat{\boldsymbol{w}}_1^\top \boldsymbol{x}_{\text{test}} + \hat{\boldsymbol{v}}_1^\top \boldsymbol{a}_{\text{test}}$. However, since $\boldsymbol{a}_{\text{test}}$ is not available in our context, we use instead its (assumed known) mean $\mu(\boldsymbol{x}_{\text{test}})$, i.e., mean imputation (Little & Rubin, 2019).

Denoting by $\boldsymbol{\Pi}_x$ the orthogonal projector associated with $\boldsymbol{X}$, defined in Appendix E, our analysis shows that as long as

- **Variance of PI**: The variance $\{(\boldsymbol{v}^\star)^\top \Sigma(\boldsymbol{x}_i)\boldsymbol{v}^\star\}_{i=1}^n$ due to PI is large enough and/or
- **Alignment of PI**: The PI features $\boldsymbol{A}$ have a significant average component outside of the subspace spanned by the features $\boldsymbol{X}$, i.e., the term below is large enough

$$\frac{1}{n}\|(\boldsymbol{I} - \boldsymbol{\Pi}_x)\mu(\boldsymbol{X})\boldsymbol{v}^\star\|^2 \qquad (1)$$

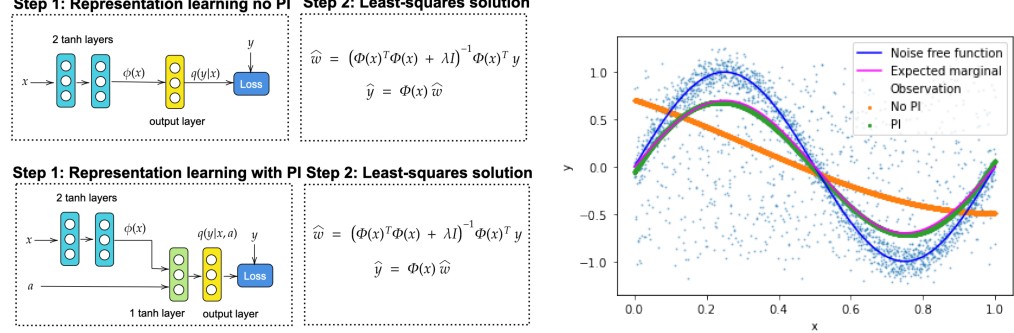

(a) Learning the representation $\phi(x)$ without PI (top) or using PI (bottom).

(b) Comparing representations learned by No PI vs. using PI in a two-step procedure.

Figure 1: Toy representation learning experiment. In Step 2 (common to both methods) we solve a least-squares problem assuming $y = [y_1, \ldots, y_N] \in \mathbb{R}^N$ and $\Phi(x) = [\phi(x_1), \ldots, \phi(x_N)] \in \mathbb{R}^{N \times r}$.

then the estimator (PI) has a lower risk compared to (NO-PI). In other words, it is provably better to exploit the privileged information $\boldsymbol{A}$ at training time instead of ignoring it.

Our analysis further covers the case of (marg.NO-PI) where we marginalize $\hat{\boldsymbol{w}}_0$ with respect to PI and we predict with $\boldsymbol{X}\mathbb{E}_{\boldsymbol{a} \sim p(\boldsymbol{a}|\boldsymbol{x})}[\hat{\boldsymbol{w}}_0]$, which we compare with (marg.PI) the marginalized predictions $\mathbb{E}_{\boldsymbol{a} \sim p(\boldsymbol{a}|\boldsymbol{x})}[\boldsymbol{X}\hat{\boldsymbol{w}}_1 + \boldsymbol{A}\hat{\boldsymbol{v}}_1]$. In that case, we can show the same conclusion as previously, with the exception that the variance term $\{(\boldsymbol{v}^\star)^\top \Sigma(\boldsymbol{x}_i)\boldsymbol{v}^\star\}_{i=1}^n$ does not have influence anymore, only (1) drives the comparison. Indeed, the proof in Appendix E shows that marginalizing removes from the risk expressions the terms related to the variance of PI.

## 2.2 PI HELPS TO LEARN BETTER REPRESENTATIONS: A MOTIVATING EXPERIMENT

Our analysis of the effect of PI in a linear model has established key insights into the conditions under which PI is provably useful. We now look at a toy non-linear learning experiment using neural networks, which provides empirical evidence that PI can be helpful in deep networks. In particular, we show that representations learned with access to useful PI can explain away label noise and transfer better than representations learned without access to PI. This motivating experiment forms the basis of our TRAM method.

We simulate a noisy annotator by assuming in this toy example that the PI is a binary indicator, $a \sim \text{Bernoulli}(0.3)$, with $a = 1$ representing the case where the noisy annotator provides a label independent of $x$:

$$y = (1 - a) \cdot \sin(2\pi x) + a \cdot v + \epsilon,$$

where $x \in [0, 1]$, $v \sim \mathcal{U}(-1, 1)$ and $\epsilon \sim \mathcal{N}(0, 0.1)$.

We then fit two networks to $N = 2,500$ training examples generated according to this process. The first network does not have access PI and is a two layer MLP, both layers of dimension 64, with tanh hidden activations and linear output activation; see top left of Fig. 1a for an illustration. The second network has access to PI, and is defined as per the two-step TRAM approach. The part of the network which learns the $x$ representation, $\phi$, is defined exactly as the no-PI MLP. $q(y|\boldsymbol{x}, \boldsymbol{a})$, the output head with access to PI, see Fig. 1a, is a single layer MLP with 64 units, tanh activation and linear output layer, with the concatenation of $a$ and $\phi(x)$ as inputs; see bottom left of Fig. 1a. Both networks are fit for 10 epochs by the Adam optimizer (Kingma & Ba, 2014) with mean squared error loss function.

We then extract the non-linear representations of $x$ learned by both networks and fit a linear model on the dataset $\{\phi(x_i), y_i\}, i = 1, \ldots, N$. The linear model can be solved exactly using the L2 regularized ordinary least squares solution. We plot the results in Fig. 1. We see that the representations learned by the model with access to PI in step #1 enable a near perfect fit to the true expected marginal distribution, $\mathbb{E}_{(\boldsymbol{a},y) \sim p(\boldsymbol{a},y|\boldsymbol{x})}[y]$ across $\mathcal{X}$ space. However *without access to PI the noise term $a \cdot v$ cannot be explained away*. As a result, the linear model fit on top of the representations learned without access to PI is substantially worse than the model fit using the two step procedure. We emphasize that both models have exactly the same capacity.

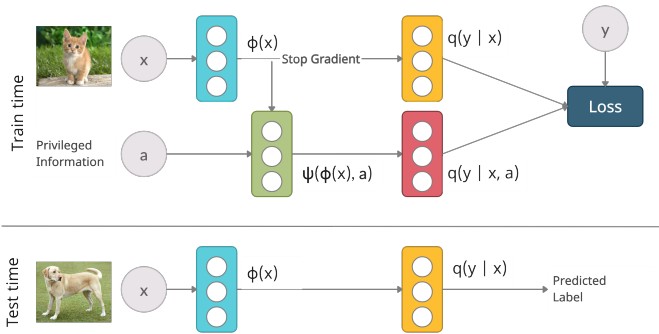

Figure 2: The TRAM method in diagrammatic form.

In Appendix F we extend this representation learning procedure to a large-scale image classification case. We learn a representation with and without access to PI on a relabeled version of ImageNet (details in §5.2) using a ResNet50 (He et al., 2016). We then freeze the representation and evaluate using a linear model. Access to PI improves the representations learned.

## 3  METHOD: TRAM

We consider learning under privileged information (Vapnik & Vashist, 2009), **LUPI**. Our proposed method, TRAM, consists of a single neural network with two output heads, providing predictions for both $p(y|\boldsymbol{x}, \boldsymbol{a})$ and $p(y|\boldsymbol{x})$; see Fig. 2. There are two key ingredients to TRAM; (i) the $p(y|\boldsymbol{x})$ head is a simple, yet a provably valid, approximation to the marginal $\int p(y|\boldsymbol{x}, \boldsymbol{a})p(\boldsymbol{a}|\boldsymbol{x})d\boldsymbol{a}$ and (ii) a partition of the parameter space such that the neural network weights are shared between the two output heads, and that these shared weights are updated solely based on the lower variance gradients from the $p(y|\boldsymbol{x}, \boldsymbol{a})$ head which has access to PI. Below we develop TRAM in the classification setting, in §3.4 we extend the method to the regression setting.

### 3.1  INGREDIENT #1: MARGINALIZE OVER PI AT TEST TIME

A natural probabilistic approach to LUPI is (i) to learn the conditional distribution $p(y|\boldsymbol{x}, \boldsymbol{a})$ during training and (ii) then, at test time, marginalize over the $\mathcal{A}$ domain, computing $p(y|\boldsymbol{x}) = \int p(y|\boldsymbol{x}, \boldsymbol{a})p(\boldsymbol{a}|\boldsymbol{x})d\boldsymbol{a}$ (Lambert et al., 2018).

Predicting with the marginal $p(y|\boldsymbol{x})$ at test time is motivated by the following key observation. Consider the set of distributions $\mathcal{Q}$ over the $C$ class labels, $\mathcal{Q} = \{q(y|\cdot)|\forall \boldsymbol{x} \in \mathcal{X}, q(y|\boldsymbol{x}) \in \Delta_C\}$ where $\Delta_C$ is the $C$-dimensional simplex. Among all the distributions $q \in \mathcal{Q}$, the marginal distribution $\boldsymbol{x} \mapsto p(y|\boldsymbol{x})$ minimizes the following optimization problem:

$$\min_{q \in \mathcal{Q}} \mathbb{E}_{(\boldsymbol{x}, \boldsymbol{a}) \sim p(\boldsymbol{x}, \boldsymbol{a})} \left[ D_{KL}(p(y|\boldsymbol{x}, \boldsymbol{a}) \, \| \, q(y|\boldsymbol{x})) \right]. \tag{2}$$

See proof in Appendix B. In words, $p(y|\boldsymbol{x})$ is optimal in the sense that it minimizes the *expected* KL divergence to $p(y|\boldsymbol{x}, \boldsymbol{a})$. Access to $p(y|\boldsymbol{x})$ enables minimizing the Bayes risk (Murphy, 2012) at test time. Note further that the mean imputation scheme which provably reduces the expected risk for a linear model, §2.1, corresponds precisely to marginalizing over PI at test time.

Directly computing $p(y|\boldsymbol{x})$ has two problems; (i) it is typically intractable and (ii) $p(\boldsymbol{a}|\boldsymbol{x})$ is unknown and therefore must be learned, which is a challenging generative modelling problem in itself, especially as $\boldsymbol{a}$ may have mixed type features. A Monte Carlo estimate of the integral using the samples from $\mathcal{A}$ in the training set is only feasible with the independence assumption $p(\boldsymbol{a}|\boldsymbol{x}) = p(\boldsymbol{a})$, so that $p(y|\boldsymbol{x})$ reduces to $\int p(y|\boldsymbol{x}, \boldsymbol{a})p(\boldsymbol{a})d\boldsymbol{a} \approx \frac{1}{S}\sum_{s=1}^{S} p(y|\boldsymbol{x}, \boldsymbol{a}_s)$ with $\boldsymbol{a}_s \sim p(\boldsymbol{a})$.

Unfortunately this independence assumption is often violated in practice. In addition the memory and computational cost of MC estimation scales linearly in $S$, the number of MC samples. This $\mathcal{O}(S)$ scaling is undesirable for production deployment with strict latency requirements.

Due to the challenge of computing the integral directly, we propose a simple approximation $q(y|\boldsymbol{x}; \boldsymbol{w})$ to $p(y|\boldsymbol{x})$. It exploits the property (2) of $p(y|\boldsymbol{x})$ as the distribution minimizing the expected KL

divergence to its conditional $p(y|\boldsymbol{x}, \boldsymbol{a})$. We choose $q$ to be cheap to evaluate at test time. For example, for a multi-class vanilla TRAM classifier $q(y|\boldsymbol{x}; \boldsymbol{w}) = \mathtt{softmax}(\boldsymbol{W}\phi(\boldsymbol{x}))$.

## 3.2 INGREDIENT #2: TRANSFER VIA WEIGHT SHARING

We partition the parameter space into four disjoint subsets;

1. Let $\phi(\boldsymbol{x})$ be a feature extractor for $\boldsymbol{x} \in \mathcal{X}$.

2. Similarly, let $\psi(\phi(\boldsymbol{x}), \boldsymbol{a})$ be a feature extractor *jointly* applied to $(\phi(\boldsymbol{x}), \boldsymbol{a})$ for $(\boldsymbol{x}, \boldsymbol{a})$ in $\mathcal{X} \times \mathcal{A}$.

3. The weights $\boldsymbol{w}$ parameterize the marginal distribution: $q(y|\boldsymbol{x}; \boldsymbol{w}) = q(y|\phi(\boldsymbol{x}); \boldsymbol{w})$.

4. The weights $\boldsymbol{u}$ parameterize the conditional distribution $q(y|\boldsymbol{x}, \boldsymbol{a}; \boldsymbol{u})$, namely $q(y|\boldsymbol{x}, \boldsymbol{a}; \boldsymbol{u}) = q(y|\psi(\phi(\boldsymbol{x}), \boldsymbol{a}); \boldsymbol{u})$.

**Two-step approach.** Motivated by Eq. (2), the connection between LUPI and multi-task learning (Jonschkowski et al., 2016) and our toy representation learning experiments, §2.2, we consider the following two-step approach:

$$\min_{\boldsymbol{u}, \phi, \psi} \mathbb{E}_{(\boldsymbol{x}, \boldsymbol{a}, y) \sim p(\boldsymbol{x}, \boldsymbol{a}, y)} \big[ \mathcal{L}_1(y, q(y|\boldsymbol{x}, \boldsymbol{a})) \big] \tag{3}$$

$$\min_{\boldsymbol{w}} \mathbb{E}_{(\boldsymbol{x}, \boldsymbol{a}, y) \sim p(\boldsymbol{x}, \boldsymbol{a}, y)} \big[ \mathcal{L}_2(y, q(y|\boldsymbol{x})) \big] \text{ with } \phi = \phi^\star \tag{4}$$

$\mathcal{L}_*$ are arbitrary loss functions. We assume $\phi$ and $\psi$ are parameterized as neural networks, so $\min_{\phi, \psi}$ refers to optimizing the network weights.

Crucially $\phi^\star$ is the feature extractor learned in (3) with access to PI. This weight sharing enables **knowledge transfer** to the network trained without PI. Given Eq. (2), we know that Eq. (4) approximates the true marginal distribution $p(y|\boldsymbol{x})$ (observe that the KL divergence in Eq. (2) reduces to the cross-entropy loss function for $\mathcal{L}_2$ when taking the one-hot training labels for $p(y|\boldsymbol{x}, \boldsymbol{a})$).

**Merging the two steps.** To further simplify the above approach, we propose to merge Eq. (3) and Eq. (4) into a *single* training procedure. To that end, and reusing the terminology commonly used in deep-learning frameworks, let us define

$$\pi(y|\boldsymbol{x}; \boldsymbol{w}) = q\big(y|\mathtt{stop\_gradient}(\phi(\boldsymbol{x})); \boldsymbol{w}\big)$$

which coincides with $q(y|\boldsymbol{x})$ except that its gradient only depends on $\boldsymbol{w}$. For some $\beta > 0$, we then consider:

$$\min_{\boldsymbol{u}, \boldsymbol{w}, \phi, \psi} \mathbb{E}_{(\boldsymbol{x}, \boldsymbol{a}, y) \sim p(\boldsymbol{x}, \boldsymbol{a}, y)} \big[ \mathcal{L}_2(y, \pi(y|\boldsymbol{x})) + \beta \mathcal{L}_1(y, q(y|\boldsymbol{x}, \boldsymbol{a})) \big]$$

as the joint training objective. In practice, since the parameters of the two losses are partitioned, we can set $\beta = 1$ and fold instead the search over $\beta$ into the search of the learning rate, hence not introducing an extra hyperparameter.

## 3.3 TRAM VARIANTS

Privileged information may only explain away some of label noise uncertainty. Below we propose two TRAM variants which combine TRAM with existing noisy labels methods.

**Het-TRAM.** Heteroscedastic classifiers are capable of modeling label noise that is input-dependent and have been successfully applied in this setting (Collier et al., 2021). Further note that even in some cases where the conditional distribution $q(y|\boldsymbol{x}, \boldsymbol{a})$ is homoscedastic, the marginal $q(y|\boldsymbol{x})$ is heteroscedastic, see Appendix C for details.

Hence we propose **Het-TRAM**, a TRAM variant in which $q(y|\boldsymbol{x})$ is heteroscedastic. This increases the expressiveness of $q$, improving the approximation in the second step of our optimization procedure, Eq. (4). We implement the method of Collier et al. (2021) to make $q(y|\boldsymbol{x})$ heteroscedastic.

**Distilled-TRAM.** Distillation (Hinton et al., 2015) is a technique for transferring knowledge between two neural networks. Distillation has been previously applied to LUPI (Lopez-Paz et al., 2015). In Distilled-TRAM we use the two step TRAM procedure, setting the loss function, $\mathcal{L}_1$ in Eq. (3), to the distillation loss. The teacher network is first trained with access to PI and in the second step the distilled-TRAM model is trained.

### 3.4 REGRESSION

We developed TRAM and Het-TRAM focussing on the classification setting but our approach is trivial to generalize to regression problems. In the regression case, we can choose the predictive distribution to be Gaussian, $q(y|\boldsymbol{x}) = \mathcal{N}(\mu(\boldsymbol{x}), \sigma^2(\boldsymbol{x}))$. For vanilla TRAM we can choose $\sigma^2(\boldsymbol{x}) = 1$, while for Het-TRAM we can choose $\sigma^2(\boldsymbol{x}) = \texttt{softplus}(\boldsymbol{w}_\sigma^\top \phi(\boldsymbol{x}))$. $\mu$ and $\sigma^2$ are parameterized as neural networks with our shared feature extractor $\phi(\boldsymbol{x})$, similar to Kendall & Gal (2017). $\mathcal{L}_1$ in Eq. (3) and $\mathcal{L}_2$ in Eq. (4) are replaced by the Gaussian negative log-likelihood.

## 4 RELATED WORK

Vapnik & Vashist (2009) develop a theoretic framework for the LUPI paradigm and introduce the SVM+ method for training Support Vector Machines in this regime. The slack variables for the SVM+ constraints are a function of the PI. SVM+ has been extended in the SVM literature (Lapin et al., 2014; Vapnik & Izmailov, 2015). Jonschkowski et al. (2016) provide a unifying framework that connects together multi-task learning, multi-view learning and LUPI.

Yang et al. (2017) extend the SVM+ approach to neural network models with their MIML-FCN+ method. The authors formulate a two-tower network similar to ours, but without weight sharing between the towers. Both towers make independent predictions given $\boldsymbol{x}$ or $\boldsymbol{a}$ as inputs. The tower with access to PI predicts the loss of the other tower and this prediction is regularized to be close to the true loss. In this way the PI tower outputs a neural network analogue to the SVM+ slack variables.

Lambert et al. (2018) utilize PI by making the training-time Gaussian-dropout variance (Kingma et al., 2015) a function of the PI. At test time the PI is approximately marginalized over by removing the dropout. Similarly Hernández-Lobato et al. (2014) allow the additive Gaussian noise component of a heteroscedastic Gaussian Process Classifier (Rasmussen & Williams, 2006) to be a function of the PI. The classifier is homoscedastic at test time.

Lopez-Paz et al. (2015) propose a distillation (Hinton et al., 2015) style approach to learning with PI. The teacher network is trained with access to PI. In the distillation step the student network is given $\boldsymbol{x}$ as input and a convex combination of soft labels from the teacher network and true labels $y$ as targets. Xu et al. (2020) extend and apply this distillation method to a recommender system.

TRAM implements knowledge transfer via weight sharing, performs efficient approximate marginalization at test time and can be applied to many widely used architectures. Lambert et al. (2018) also share weights and approximate the marginal $p(y|\boldsymbol{x})$ however they require the use of Gaussian dropout, which is not widely used. The distillation and MIML-FCN+ methods do not transfer via weight sharing and do not approximate $p(y|\boldsymbol{x})$. Distillation also requires a two-step training procedure. See Table 5, in Appendix F for a comparison of the key features of selected LUPI methods.

## 5 EXPERIMENTS

Our experiments tackle the general LUPI problem under label noise. There are a few large-scale public datasets with PI We thus use both real-world datasets with PI as well as synthesizing PI for a re-labelled version of ImageNet (Deng et al., 2009).

We evaluate a number of baselines in addition to our method.

- The "No PI" baseline is standard neural network training which directly learns $p(y|\boldsymbol{x})$ and never uses PI.
- Zero and mean imputation learn $p(y|\boldsymbol{x}, \boldsymbol{a})$ at training time and substitute $\boldsymbol{a} = \boldsymbol{0}$ and $\boldsymbol{a} = \frac{1}{N}\sum_i \boldsymbol{a}_i$ respectively at test time. For mean imputation, averaging takes place after feature pre-processing, e.g., one-hot encoding of the annotator ID.

Table 1: CIFAR-10 negative log-likelihood & accuracy (trained on CIFAR-10H). Averaged over 20 training runs $\pm$ 1 std. dev.

| METHOD | ↓NLL | ↑ACCURACY |
|---|---|---|
| NO PI | $1.058 \pm 0.050$ | $67.0 \pm 1.7$ |
| ZERO IMPUTATION | $1.009 \pm 0.032$ | $68.7 \pm 1.4$ |
| MEAN IMPUTATION | $\mathbf{0.963} \pm 0.058$ | $70.1 \pm 1.5$ |
| LAMBERT ET AL. (2018) | $1.033 \pm 0.044$ | $67.1 \pm 1.3$ |
| FULL MARGINALIZATION | $1.119 \pm 0.058$ | $70.3 \pm 2.5$ |
| TRAM | $0.980 \pm 0.037$ | $70.1 \pm 1.4$ |
| HET-TRAM | $0.972 \pm 0.038$ | $\mathbf{70.4} \pm 1.5$ |
| DISTILLATION NO PI | $1.118 \pm 0.037$ | $70.1 \pm 1.4$ |
| LOPEZ-PAZ ET AL. (2015) | $1.121 \pm 0.040$ | $70.2 \pm 1.4$ |
| DISTILLED-TRAM | $\mathbf{0.941} \pm 0.039$ | $\mathbf{71.8} \pm 1.4$ |

- The "Full marginalization" baseline is an expensive MC estimate of $p(y|\boldsymbol{x}) = \int p(y|\boldsymbol{x}, \boldsymbol{a})p(\boldsymbol{a}|\boldsymbol{x})d\boldsymbol{a}$ at test time, see §3.1 for details. It is a gold standard (up to independence assumption error), impractical to compute in many applications.

- We also compare against distillation based approaches. "Distillation No PI" is an ablation to see the effect of distillation alone, independent of PI, in which a network trained *without* access to PI is distilled into another network *also without* access to PI.

Prior work did not evaluate against these simple imputation baselines or full marginalization (Lopez-Paz et al., 2015; Yang et al., 2017; Lambert et al., 2018).

## 5.1 CIFAR-10H

One dataset with annotator features is CIFAR-10H (Peterson et al., 2019), which is a re-labelled version of the CIFAR-10 (Krizhevsky & Hinton, 2009) test set. The new labels are provided by crowd-sourced human annotators. We make use of three annotator features; the annotator ID, the reaction time of the annotator to provide the label and how much experience the annotator had with the task, as measured by the number of labels the annotator had previously provided.

As we only have annotator features for the CIFAR-10 test set, we use this as our training set and evaluate on the official training set. As a result we have only 10,000 images for training. To achieve reasonable performance we start from a MobileNet (Howard et al., 2017) pretrained on ImageNet. Images have on average $> 50$ annotations each. This is unrealistic for typical applications where 1-3 labels per example is more common. Therefore, we subsample 16,400 labels (1.64 labels per example), see Appendix D for details of the subsampling procedure. The subsampled labels agree with the true CIFAR-10 test set labels 79.4% of the time.

In Table 1 we see the results. First, and as expected, using annotator features via TRAM, marginalization or the imputation methods provides a performance improvement over standard neural network training without PI. Second, we see that TRAM performs on par with full marginalization (which uses 16,400 MC samples of $\boldsymbol{a}$ from the training set), despite having constant time compute and memory requirements w.r.t. the number of MC samples for the full marginalization baseline. Mean imputation is a strong baseline on CIFAR-10H. Het-TRAM improves over TRAM demonstrating the efficacy of making $q(y|\boldsymbol{x}, \boldsymbol{a})$ heteroscedastic. It is noteworthy that distillation using PI, (Lopez-Paz et al., 2015) does not improve over standard distillation without PI. However Distilled-TRAM with makes use of PI for distillation but then performs approximate marginalization and transfer learning via weight sharing improves over the distillation baselines on both accuracy and log-likelihood metrics.

### 5.1.1 QUALITATIVE ANALYSIS OF CIFAR-10H RESULTS

We qualitatively analyse how PI is helping improve the performance of TRAM on CIFAR-10H. The PI for CIFAR-10H does not contain a feature for annotator reliability. However the PI does contain features such as the annotator ID and the annotator reaction time from which it may be possible to

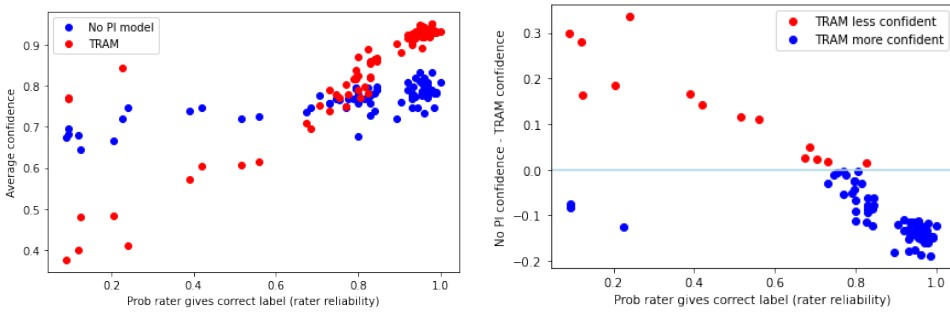

(a) Average confidence per model.

(b) Delta in average confidence between models.

Figure 3: How model confidence varies with annotator reliability for CIFAR-10H. Each point represents a single human annotator. The x-value is the probability the annotator's label agrees with the true CIFAR-10 label. See individual captions for y-value meaning. Confidence is defined as the maximum probability given by the model across the 10 labels.

Table 2: ImageNet validation neg-log-likelihood and accuracy. Avg. over 10 seeds $\pm$ 1 std. dev.

| METHOD | ↓NLL | ↑ACCURACY |
|---|---|---|
| NO PI | $1.264 \pm 0.007$ | $71.7 \pm 0.2$ |
| ZERO IMPUTATION | $1.895 \pm 0.008$ | $63.5 \pm 0.2$ |
| MEAN IMPUTATION | $1.619 \pm 0.007$ | $65.1 \pm 0.3$ |
| LAMBERT ET AL. (2018) | $1.264 \pm 0.006$ | $71.8 \pm 0.1$ |
| FULL MARGINALIZATION | $1.217 \pm 0.004$ | $72.6 \pm 0.2$ |
| TRAM | $1.225 \pm 0.006$ | $72.5 \pm 0.2$ |
| HET-TRAM | $\mathbf{1.207} \pm 0.008$ | $\mathbf{72.8} \pm 0.2$ |
| DISTILLATION NO PI | $1.207 \pm 0.004$ | $72.6 \pm 0.2$ |
| LOPEZ-PAZ ET AL. (2015) | $1.216 \pm 0.003$ | $72.7 \pm 0.2$ |
| DISTILLED-TRAM | $\mathbf{1.154} \pm 0.004$ | $\mathbf{73.8} \pm 0.2$ |

learn to trust some annotators more than others. TRAM can *learn* output a less confident distribution for unreliable annotators, thus reducing the harmful impact of incorrect labels.

In Fig. 3 we show an analysis of the confidence of the TRAM and No PI (i.e., standard) models for each annotator in CIFAR-10H. We see that the trend for TRAM is a strong linear relationship between the reliability of an annotator and the confidence of the model, Fig. 3a. The TRAM model is consistently more confident than the No PI model for reliable annotators while the No PI model is overconfident for unreliable annotators, Fig. 3b

## 5.2 IMAGENET ILSVRC12

In order to create a large-scale dataset with annotator features, we re-label the ImageNet ILSVRC12 training set by the following procedure. We download 16 different models pre-trained on ImageNet, see Appendix D for further details. We also add a 17th malicious annotator model which picks a label uniformly at random from the 1,000 ImageNet ILSVRC12 classes. For each image in the training set we select the malicious annotator with 10% probability and otherwise sample one of the 16 models with equal probability. We then sample a label from the predictive distribution of that model for that image. This is the label used for training. On average the sampled label agrees with the true ImageNet label 68.3% of the time. The annotator features are the model ID (a proxy for a human annotator ID) and the probability of the label assigned by the model (a proxy for the confidence of a human annotator). The ImageNet image is used as the non-privileged information $\boldsymbol{x}$. $\phi$ is randomly initialized ResNet-50 (He et al., 2016).

See Table 2 for the results. The full marginalization baseline uses 1,000 MC samples of $\boldsymbol{a}$ from the training set. The imputation baselines perform worse than not using PI, perhaps due to the imputed values having low density $p(\boldsymbol{a}|\boldsymbol{x})$. Again TRAM performs on par with full marginalization and Het-TRAM has higher accuracy than both. The ImageNet labels are known to exhibit heteroscedasticity

Table 3: Civil Comments Identities test set negative log-likelihood and average accuracy over 7 classes. Averaged over 10 training runs $\pm$ 1 std. dev.

| METHOD | ↓NLL | ↑ AVG. ACCURACY |
|---|---|---|
| NO PI | $0.085 \pm 0.011$ | $97.8 \pm 0.12$ |
| ZERO IMPUTATION | $0.073 \pm 0.004$ | $\mathbf{98.2} \pm 0.01$ |
| MEAN IMPUTATION | $0.069 \pm 0.003$ | $\mathbf{98.2} \pm 0.02$ |
| LAMBERT ET AL. (2018) | $0.084 \pm 0.012$ | $97.8 \pm 0.17$ |
| FULL MARGINALIZATION | $0.065 \pm 0.004$ | $97.8 \pm 0.00$ |
| TRAM | $0.064 \pm 0.002$ | $\mathbf{98.2} \pm 0.01$ |
| HET-TRAM | $\mathbf{0.062} \pm 0.001$ | $98.1 \pm 0.1$ |
| DISTILLATION NO PI | $0.094 \pm 0.011$ | $97.8 \pm 0.1$ |
| LOPEZ-PAZ ET AL. (2015) | $0.089 \pm 0.000$ | $97.8 \pm 0.0$ |
| DISTILLED-TRAM | $0.065 \pm 0.001$ | $\mathbf{98.2} \pm 0.0$ |

(Collier et al., 2021), therefore we make both $q(y|\boldsymbol{x})$ and $q(y|\boldsymbol{x}, \boldsymbol{a})$ heads heteroscedastic for Het-TRAM. Distilled-TRAM has significantly better NLL and accuracy than the two distillation baselines.

## 5.3 CIVIL COMMENTS

We further evaluate our method on a large-scale text classification dataset. Civil Comments[1] is a collection of comments from independent news websites annotated with 7 toxicity labels (identity attack, insult, obscene, severe toxicity, sexually explicit, threat, toxicity). The Civil Comments Identities subset of the Civil Comments data contains privileged information in the form of 24 attributes identified in the comment (male, female, christian and so on). The Identities subset consists of 405,130 training examples, 21,293 validation examples and 21,577 test set examples.

The shared network $\phi$ is a pre-trained Universal Sentence Encoder (Cer et al., 2018). Table 3 contains the test set results. We report negative log-likelihood and accuracy averaged over the 7 labels. The TRAM, Het-TRAM and imputation methods perform similarly well in terms of average accuracy, outperforming the No PI baseline as well as the Gaussian Dropout and full marginalization methods.

The poor accuracy of the full marginalization method is interesting to note. The PI is directly derived from the non-PI (in the form of 24 identity human labelled attributes for the non-PI). This is clearly a violation of the independence assumption required for a MC estimate of full marginalization to be computable. The dependence of $\boldsymbol{a}$ on $\boldsymbol{x}$ is most clearly identifiable for the Civil Comments Identities dataset; as a result the relative performance of the full marginalization method is poorest on this dataset. Further note that the TRAM and Het-TRAM methods have lower negative log-likelihood than all other baseline methods. Standard distillation with no PI and Lopez-Paz et al. (2015) style distillation where the teacher network is trained with PI does not provide a performance improvement over the no PI baseline. Distilled-TRAM performs on par with vanilla TRAM.

## 6 CONCLUSION

We introduced TRAM, a new method for LUPI in supervised neural networks. TRAM (i) learns an efficient, simple distribution to approximately marginalize over PI at test time and (ii) partitions the parameter space enabling transfer via weight sharing of the knowledge learned with access to PI. TRAM can be successfully combined with established methods for dealing with noisy labels; distillation (Distilled-TRAM) and heteroscedastic output layers (Het-TRAM). We have analysed a linear model with PI where deriving analytic results are feasible. In this setting we have shown the utility of using PI and ingredient #1 of our TRAM method, the marginalization over PI. Using a toy low-dimensional problem we have further shown the effectiveness of ingredient #2 of our proposed TRAM method, transfer learning via weight sharing of representations learned with access to PI. We then have empirically validated the single-step TRAM procedure on larger-scale datasets in the image and text domain; CIFAR-10H, a noisy version of ImageNet and Civil Comments Identities.

---

[1] https://www.kaggle.com/c/jigsaw-unintended-bias-in-toxicity-classification/data

## 7 ETHICS AND REPRODUCIBILITY STATEMENTS

**Ethics.** We present a generic algorithm for classification/regression with privileged information. The method is not application specific and has minimal computational overhead. We do not see ethical issues specific to our method beyond any already associated with LUPI methods and/or deep neural network classifiers.

**Reproducibility.** We provide detailed information on the data generation process for the CIFAR-10H and Imagenet datasets in Appendix D.1. Hyperparameters and further details required to reproduce our results are provided in Appendix D.2. We plan to release publicly Colab notebooks to reproduce our toy transfer learning and Civil Comments experiments. We also plan to make public our Imagenet dataset in Tensorflow Dataset format to be used as a large-scale dataset with annotator features by researchers working on the LUPI problem. Detailed derivations with conditions for our theoretical analysis are presented in Appendix E.

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

## A  APPENDIX

## B  PROOF OF EQUATION (2)

As a reminder, we consider $C$ class labels and denote by $\Delta_C$ the $C$-dimensional simplex. We define the set of distributions $\mathcal{Q}$ over the $C$ class labels by

$$\mathcal{Q} = \{q(y|\cdot) \,|\, \forall \boldsymbol{x} \in \mathcal{X}, q(y|\boldsymbol{x}) \in \Delta_C\}.$$

Consider the optimization problem

$$\min_{q \in \mathcal{Q}} \mathbb{E}_{\boldsymbol{x} \sim p(\boldsymbol{x})} \left[ D_{KL}(p(y|\boldsymbol{x}) \,\|\, q(y|\boldsymbol{x})) \right] \tag{5}$$

whose solution is straightforwardly given by the marginal distribution $\boldsymbol{x} \mapsto q^{\star}(y|\boldsymbol{x}) = p(y|\boldsymbol{x})$. We recall that the KL $D_{KL}(p(y|\boldsymbol{x}) \,\|\, q(y|\boldsymbol{x}))$ is defined by

$$D_{KL}(p(y|\boldsymbol{x}) \,\|\, q(y|\boldsymbol{x})) = \sum_{j=1}^{C} p_j(y|\boldsymbol{x}) \log \left( \frac{p_j(y|\boldsymbol{x})}{q_j(y|\boldsymbol{x})} \right). \tag{6}$$

For any $\boldsymbol{x}$ and $j \leq C$, we can rewrite the terms of the sum

$$p_j(y|\boldsymbol{x}) \log \left( \frac{p_j(y|\boldsymbol{x})}{q_j(y|\boldsymbol{x})} \right)$$

as

$$\mathbb{E}_{\boldsymbol{a}|\boldsymbol{x} \sim p(\boldsymbol{a}|\boldsymbol{x})} \left[ p_j(y|\boldsymbol{x}, \boldsymbol{a}) \log \left( \frac{p_j(y|\boldsymbol{x})}{q_j(y|\boldsymbol{x})} \right) \right]$$

where we have used (i) the fact that $\log(\frac{p_j(y|\boldsymbol{x})}{q_j(y|\boldsymbol{x})})$ does not depend on $\boldsymbol{a}$ and (ii) the definition of the marginal distribution

$$\begin{aligned} p_j(y|\boldsymbol{x}) &= \int p_j(y|\boldsymbol{x}, \boldsymbol{a}) p(\boldsymbol{a}|\boldsymbol{x}) d\boldsymbol{a} \\ &= \mathbb{E}_{\boldsymbol{a}|\boldsymbol{x} \sim p(\boldsymbol{a}|\boldsymbol{x})} \left[ p_j(y|\boldsymbol{x}, \boldsymbol{a}) \right]. \end{aligned}$$

Multiplying and dividing in the argument of the log by $p_j(y|\boldsymbol{x}, \boldsymbol{a})$, we obtain

$$\mathbb{E}_{\boldsymbol{a}|\boldsymbol{x} \sim p(\boldsymbol{a}|\boldsymbol{x})} \left[ p_j(y|\boldsymbol{x}, \boldsymbol{a}) \log \left( \frac{p_j(y|\boldsymbol{x}, \boldsymbol{a})}{q_j(y|\boldsymbol{x})} \frac{p_j(y|\boldsymbol{x})}{p_j(y|\boldsymbol{x}, \boldsymbol{a})} \right) \right].$$

Summing over $j \in \{1, \ldots, C\}$ to reconstruct the KL term (6), this leads to, for any $\boldsymbol{x}$,

$$\begin{aligned} D_{KL}(p(y|\boldsymbol{x}) \,\|\, q(y|\boldsymbol{x})) &= \mathbb{E}_{\boldsymbol{a}|\boldsymbol{x}} \left[ D_{KL}(p(y|\boldsymbol{x}, \boldsymbol{a}) \,\|\, q(y|\boldsymbol{x})) \right] \\ &\quad - \mathbb{E}_{\boldsymbol{a}|\boldsymbol{x}} \left[ D_{KL}(p(y|\boldsymbol{x}, \boldsymbol{a}) \,\|\, p(y|\boldsymbol{x})) \right]. \end{aligned}$$

Since the second term above *does not depend on $q$*, minimizing (5) is equivalent to minimizing

$$\begin{aligned} &\min_{q \in \mathcal{Q}} \quad \mathbb{E}_{\boldsymbol{x}} \left[ \mathbb{E}_{\boldsymbol{a}|\boldsymbol{x}} \left[ D_{KL}(p(y|\boldsymbol{x}, \boldsymbol{a}) \,\|\, q(y|\boldsymbol{x})) \right] \right] \\ =\ &\min_{q \in \mathcal{Q}} \quad \mathbb{E}_{(\boldsymbol{x}, \boldsymbol{a}) \sim p(\boldsymbol{x}, \boldsymbol{a})} \left[ D_{KL}(p(y|\boldsymbol{x}, \boldsymbol{a}) \,\|\, q(y|\boldsymbol{x})) \right] \end{aligned}$$

which is equal to (2) and which is, analogously to (5), minimized by the marginal distribution $\boldsymbol{x} \mapsto q^{\star}(y|\boldsymbol{x}) = p(y|\boldsymbol{x})$.

## C  HETEROSCEDASTIC MOTIVATION

We consider a simplified special case of our framework in which the conditional model $p(y|\boldsymbol{x}, \boldsymbol{a})$ is *homoscedastic* but the optimal variational distribution in the sense of Eq. 2 is *heteroscedastic*. This motivates **Het-TRAM**, in which the variational approximations $q(y|\boldsymbol{x})$ and $q(y|\boldsymbol{x}, \boldsymbol{a})$ are heteroscedastic.

Suppose we have a regression dataset constructed from labels assigned by $M$ annotators. Each annotator has their own homoscedastic Gaussian model $p(y|\boldsymbol{x}, a = m) = \mathcal{N}(\mu_{\theta_m}(\boldsymbol{x}), 1)$. Here the

Table 4: Pre-trained models used to re-label ImageNet ILSVRC12 training set and their accuracy on that training set.

| Model | Training set accuracy |
|---|---|
| ResNet50V2 | 0.70086 |
| ResNet101V2 | 0.72346 |
| ResNet152V2 | 0.72738 |
| DenseNet121 | 0.74782 |
| DenseNet169 | 0.76184 |
| DenseNet201 | 0.77344 |
| InceptionResNetV2 | 0.8049 |
| InceptionV3 | 0.77994 |
| MobileNet | 0.70594 |
| MobileNetV2 | 0.71458 |
| MobileNetV3Large | 0.75622 |
| MobileNetV3Small | 0.68158 |
| NASNetMobile | 0.74302 |
| VGG16 | 0.71178 |
| VGG19 | 0.71156 |
| Xception | 0.79076 |

PI is a single discrete Categorical feature representing the annotator ID which takes one of $M$ values with equal probability, $a \sim \text{Cat}(\frac{1}{M})$.

The marginal $p(y|\boldsymbol{x})$ is a Gaussian Mixture Model. We choose our variational family to be the Gaussian distribution, $q(y|\boldsymbol{x}) = \mathcal{N}(\mu(\boldsymbol{x}), \sigma^2(\boldsymbol{x}))$. The values of $\mu$ and $\sigma^2$ that minimize Eq. 2 are: $\mu_*(\boldsymbol{x}) = \frac{1}{M}\sum_m \mu_{\theta_m}(\boldsymbol{x})$ and $\sigma_*^2(\boldsymbol{x}) = (M - \mu_*(\boldsymbol{x})) + \frac{1}{M}\sum_m \mu_{\theta_m}^2(\boldsymbol{x})$ (Lakshminarayanan et al., 2017). Crucially note that despite the conditional distribution being homoscedastic, the best variational distribution is heteroscedastic as the variance depends on the location in $\mathcal{X}$ space.

## D EXPERIMENTAL DETAILS

### D.1 DATA GENERATION PROCESS

**CIFAR-10H.** We use the CIFAR-10 image as the non-privileged information $\boldsymbol{x}$. The annotator ID, the number of prior annotations the annotator has provided and the reaction time in milliseconds of the annotator, are used as privileged information $\boldsymbol{a}$. For feature pre-processing the annotator ID is one-hot encoded. The number of prior annotations and the reaction time are independently divided into 10 equally sized quantiles and the quantile ID is one-hot encoded. The image is pre-processed according the the standard MobileNet pre-processing (Howard et al., 2017).

As CIFAR-10H has on average more than 50 annotations per image and the labels are not particularly noisy. We subsample the CIFAR-10H labels by the following procedure. We keep all labels by the 41 annotators that agree with the true CIFAR-10 label less than 85% of the time. We then select a further 41 annotators from the remaining annotators. The average agreement of the bad annotators with the CIFAR-10 label is 63.3%, in the full subset of labels: 79.2% and in the full CIFAR-10H dataset: 94.9%. The subsampling procedure leaves 16,400 labels from 82 annotators while the full CIFAR-10H dataset has 514,200 labels from 2,571 annotators.

**ImageNet.** The annotator features are the model ID used to re-label $\boldsymbol{x}$, which is one-hot encoded and the probability of that label being sampled. See main paper for details on the sampling procedure and see Table 4 for the list of models used and their accuracy on the ImageNet training set. The pre-trained models are downloaded from tf.keras.applications[2].

---

[2] https://www.tensorflow.org/api_docs/python/tf/keras/applications

### D.2 HYPERPARAMETERS

**CIFAR-10H.** For all methods $\phi(\boldsymbol{x})$ (or equivalent) is a MobileNet (Howard et al., 2017) pre-trained on ImageNet ILSVRC12, followed by a global average pooling layer and a Dense + ReLU layer with 64 units. $\psi(\boldsymbol{x}, \boldsymbol{a})$ is a two-layer MLP with 64 units per layer and ReLU activation. The first layer takes only $\boldsymbol{a}$ as an input, while the second layer takes the output of the first layer concatenated with $\phi(\boldsymbol{x})$ as input.

All models are trained for 20 epochs with the Adam optimizer with base learning rate= 0.001, $\beta_1 = 0.9$, $\beta_2 = 0.999$, $\epsilon = 1e - 07$. All models are trained with L2 weight regularization with weighting $1e - 3$.

Heteroscedastic models are trained using the method of Collier et al. (2021) with 4 factors for the low-rank covariance matrix approximation and a softmax temperature parameter of $\tau = 3.0$. Distilled models are also trained with a softmax temperature of $\tau = 3.0$ to smooth the teacher labels and with the distillation hyperparameter $\lambda = 0.5$ which weights the losses from the soft teacher labels and the true labels. A grid search over $\tau \in \{1.0, 2.0, 3.0, 4.0\}$ and $\lambda \in \{0.0, 0.25, 0.5, 0.75, 1.0\}$ was conducted.

**ImageNet.** For all methods $\phi(\boldsymbol{x})$ (or equivalent) is a randomly initialized ResNet-50 (He et al., 2016) with the output layer removed. $\psi(\boldsymbol{x}, \boldsymbol{a})$ is a two-layer MLP with 128 units per layer and ReLU activation, the output of this MLP is concatenated with $\phi(\boldsymbol{x})$ and then passed to the output layer. The first layer of the $\psi(\boldsymbol{x}, \boldsymbol{a})$ MLP takes only $\boldsymbol{a}$ as an input, while the second layer takes the output of the first layer concatenated with $\phi(\boldsymbol{x})$ as input.

All but Het-TRAM models are trained for 90 epochs with the SGD optimizer with base learning rate= 0.1, decayed by a factor of 10 after 30, 60 and 80 epochs. Following Collier et al. (2021), Het-TRAM is trained for 270 epochs with the same initial learning rate and learning rate decay at 90, 180 and 240 epochs. All models are trained with L2 weight regularization with weighting $1e - 4$.

Heteroscedastic models use 15 factors for the low-rank covariance matrix approximation and a softmax temperature parameter of $\tau = 1.5$. Distilled models are trained with a softmax temperature of $\tau = 3.0$ and with the distillation hyperparameter $\lambda = 0.5$. A grid search over $\tau \in \{1.0, 2.0, 3.0, 4.0\}$ and $\lambda \in \{0.0, 0.25, 0.5, 0.75, 1.0\}$ was conducted.

## E  RISK ANALYSIS

**Generative model and notations.** We assume the following

- $\boldsymbol{a} \in \mathbb{R}^m, \boldsymbol{x} \in \mathbb{R}^d$,
- $\boldsymbol{a} \sim p(\boldsymbol{a}|\boldsymbol{x}) = \mathcal{N}(\mu(\boldsymbol{x})|\Sigma(\boldsymbol{x}))$ for some mean and covariance dependent on $\boldsymbol{x}$,
- $y = \boldsymbol{x}^\top \boldsymbol{w}^\star + \boldsymbol{a}^\top \boldsymbol{v}^\star + \varepsilon$ with $\varepsilon \sim \mathcal{N}(0, \sigma^2)$.

When considering $n$ observations from this generative model, we use the matrix representations $\boldsymbol{y} \in \mathbb{R}^n, \boldsymbol{X} \in \mathbb{R}^{n \times d}, \boldsymbol{A} \in \mathbb{R}^{n \times m}$ and $\varepsilon \in \mathbb{R}^n$. We also write the zero-mean Gaussian vector

$$\boldsymbol{z} = (\boldsymbol{A} - \mu(\boldsymbol{X}))\boldsymbol{v}^\star + \varepsilon \in \mathbb{R}^n \sim \mathcal{N}(\boldsymbol{0}, \sigma^2 \boldsymbol{I} + \boldsymbol{\Lambda})$$

where we have defined the diagonal covariance

$$\boldsymbol{\Lambda} = \boldsymbol{\Lambda}(\boldsymbol{v}^\star, \boldsymbol{X}) = \text{Diag}\big(\{(\boldsymbol{v}^\star)^\top \Sigma(\boldsymbol{x}_i) \boldsymbol{v}^\star\}_{i=1}^n\big) \in \mathbb{R}^{n \times n}.$$

We list below some notation that we will repeatedly use

- The orthogonal projector associated with $\boldsymbol{X}$:

$$\boldsymbol{\Pi}_x = \boldsymbol{X}(\boldsymbol{X}^\top \boldsymbol{X})^{-1} \boldsymbol{X}^\top \in \mathbb{R}^{n \times n}.$$

- Similarly, the orthogonal projector associated with $\boldsymbol{A}$:

$$\boldsymbol{\Pi}_a = \boldsymbol{A}(\boldsymbol{A}^\top \boldsymbol{A})^{-1} \boldsymbol{A}^\top \in \mathbb{R}^{n \times n}.$$

- The projections $\boldsymbol{X}_{a\perp} = (\boldsymbol{I} - \boldsymbol{\Pi}_a)\boldsymbol{X}$ and $\boldsymbol{A}_{x\perp} = (\boldsymbol{I} - \boldsymbol{\Pi}_x)\boldsymbol{A}$.
- The matrices: $\boldsymbol{H} = (\boldsymbol{X}^\top \boldsymbol{X})^{-1}\boldsymbol{X}^\top \in \mathbb{R}^{d \times n}$ and $\boldsymbol{G} = (\boldsymbol{A}^\top \boldsymbol{A})^{-1}\boldsymbol{A}^\top \in \mathbb{R}^{m \times n}$.
- The matrices above when restricted to the projections of $\boldsymbol{X}$ and $\boldsymbol{A}$ respectively, that is,
$$\boldsymbol{H}_{a\perp} = (\boldsymbol{X}_{a\perp}^\top \boldsymbol{X}_{a\perp})^{-1}\boldsymbol{X}_{a\perp}^\top \in \mathbb{R}^{d\times n} \;\; \text{and} \;\; \boldsymbol{G}_{x\perp} = (\boldsymbol{A}_{x\perp}^\top \boldsymbol{A}_{x\perp})^{-1}\boldsymbol{A}_{x\perp}^\top \in \mathbb{R}^{m\times n}.$$

### E.1 Definition of the risk

We will compare different estimators based on their different *risks*. We focus on the *fixed* design analysis (Bach, 2021), i.e., we study the errors only due to resampling the noise $\varepsilon$ and the feature $\boldsymbol{a}$.

Given a predictor $\tau(\boldsymbol{X})$ based on the training quantities $(\boldsymbol{X}, \boldsymbol{A}, \varepsilon)$, we consider $\boldsymbol{y}' = \boldsymbol{X}\boldsymbol{w}^\star + \boldsymbol{A}'\boldsymbol{v}^\star + \varepsilon'$ (where the prime is to stress the difference with the training quantities without prime) and define the risk of $\tau$ as

$$\mathcal{R}(\tau(\boldsymbol{X}))) = \mathbb{E}_{\varepsilon' \sim p(\varepsilon'), \boldsymbol{a}' \sim p(\boldsymbol{a}'|\boldsymbol{x})}\left\{\frac{1}{n}\|\boldsymbol{y}' - \tau(\boldsymbol{X})\|^2\right\}. \tag{7}$$

Expanding the square with $\boldsymbol{y}' - \tau(\boldsymbol{X}) = \boldsymbol{X}\boldsymbol{w}^\star - \tau(\boldsymbol{X}) + \mu(\boldsymbol{X})\boldsymbol{v}^\star + \boldsymbol{z}'$, we obtain the expression

$$\mathcal{R}(\tau(\boldsymbol{X})) = \frac{1}{n}\|\boldsymbol{X}\boldsymbol{w}^\star - \tau(\boldsymbol{X}) + \mu(\boldsymbol{X})\boldsymbol{v}^\star\|^2 + \frac{1}{n}\mathrm{tr}(\sigma^2 \boldsymbol{I} + \boldsymbol{\Lambda}). \tag{8}$$

Following common practices (Bach, 2021), to assess the risk, we finally take a second expectation $\mathbb{E}_{\varepsilon \sim p(\varepsilon), \boldsymbol{a} \sim p(\boldsymbol{a}|\boldsymbol{x})}[\mathcal{R}(\tau(\boldsymbol{X}))]$ with respect to the training quantities $(\boldsymbol{A}, \varepsilon)$.

Since we will mostly consider differences of risks, we omit the variance term $\frac{1}{n}\mathrm{tr}(\sigma^2 \boldsymbol{I} + \boldsymbol{\Lambda})$ in the equations below.

### E.2 Capturing the benefit of PI without marginalization

We first describe when, in absence of any marginalization, ordinary least squares ignoring PI is worse than ordinary least squares using PI with mean imputation at prediction time.

**Proposition E.1.** *Assume that $\boldsymbol{X}^\top \boldsymbol{X}$ is invertible. Moreover, assume that $\boldsymbol{A}^\top \boldsymbol{A}$ and $[\boldsymbol{X}, \boldsymbol{A}]^\top [\boldsymbol{X}, \boldsymbol{A}]$ are almost surely invertible. We have that*

$$\mathbb{E}[\mathcal{R}(\tau_{\text{NO-PI}}(\boldsymbol{X}))] > \mathbb{E}[\mathcal{R}(\tau_{\text{PI}}(\boldsymbol{X}))]$$

*if and only if*

$$\frac{1}{n}\|(\boldsymbol{I} - \boldsymbol{\Pi}_x)\mu(\boldsymbol{X})\boldsymbol{v}^\star\|^2 + \frac{\sigma^2 d}{n} + \frac{1}{n}tr(\boldsymbol{\Pi}_x \boldsymbol{\Lambda}) > \frac{\sigma^2}{n}\mathbb{E}[\|\boldsymbol{K}\|^2]$$

*with $\boldsymbol{K} = \boldsymbol{X}\boldsymbol{H}_{a\perp} + \mu(\boldsymbol{X})\boldsymbol{G}_{x\perp}$. When $m = 1$ (i.e., $\boldsymbol{A}$ is a column vector), a sufficient condition is*

$$\frac{1}{n}\|(\boldsymbol{I} - \boldsymbol{\Pi}_x)\mu(\boldsymbol{X})\boldsymbol{v}^\star\|^2 + \frac{1}{n}tr(\boldsymbol{\Pi}_x \boldsymbol{\Lambda}) > 2\mathbb{E}\left[\frac{\|\boldsymbol{\Pi}_x \boldsymbol{A}\|^2 + \|\mu(\boldsymbol{X})\|^2}{\|(\boldsymbol{I} - \boldsymbol{\Pi}_x)\boldsymbol{A}\|^2}\right] + \frac{\sigma^2 d}{n}.$$

We provide the details of the derivation of the risk for $\tau_{\text{NO-PI}}$ and $\tau_{\text{PI}}$ in Section E.2.1 and Section E.2.2 respectively. Moreover, the second part of the proposition follows from an application of Lemma E.5.

#### E.2.1 Ordinary least squares (no marginalization)

The solution of

$$\min_{\boldsymbol{w}} \frac{1}{2}\|\boldsymbol{y} - \boldsymbol{X}\boldsymbol{w}\|^2$$

is given by $\hat{\boldsymbol{w}}_0 = (\boldsymbol{X}^\top \boldsymbol{X})^{-1}\boldsymbol{X}^\top \boldsymbol{y} = \boldsymbol{H}\boldsymbol{y}$. The corresponding predictions are

$$\tau_{\text{NO-PI}}(\boldsymbol{X}) = \boldsymbol{X}\hat{\boldsymbol{w}}_0 = \boldsymbol{\Pi}_x \boldsymbol{y} = \boldsymbol{X}\boldsymbol{w}^\star + \boldsymbol{\Pi}_x \mu(\boldsymbol{X})\boldsymbol{v}^\star + \boldsymbol{\Pi}_x \boldsymbol{z}.$$

Plugging into (8), we obtain

$$\mathcal{R}(\tau_{\text{NO-PI}}(\boldsymbol{X})) = \frac{1}{n}\|(\boldsymbol{I} - \boldsymbol{\Pi}_x)\mu(\boldsymbol{X})\boldsymbol{v}^\star - \boldsymbol{\Pi}_x \boldsymbol{z}\|^2.$$

Expanding the square and using that $\mathrm{tr}(\boldsymbol{\Pi}_x) = d$, the final risk expression is

$$\begin{aligned}
\mathbb{E}[\mathcal{R}(\tau_{\text{NO-PI}}(\boldsymbol{X}))] &= \frac{1}{n}\|(\boldsymbol{I} - \boldsymbol{\Pi}_x)\mu(\boldsymbol{X})\boldsymbol{v}^\star\|^2 + \frac{1}{n}\mathbb{E}[\|\boldsymbol{\Pi}_x \boldsymbol{z}\|^2] \\
&= \frac{1}{n}\|(\boldsymbol{I} - \boldsymbol{\Pi}_x)\mu(\boldsymbol{X})\boldsymbol{v}^\star\|^2 + \frac{\sigma^2 d}{n} + \frac{1}{n}\mathrm{tr}(\boldsymbol{\Pi}_x \boldsymbol{\Lambda}).
\end{aligned} \tag{9}$$

### E.2.2 ORDINARY LEAST SQUARES WITH PI AND MEAN IMPUTATION (NO MARGINALIZATION)

We focus on the solution of

$$\min_{\boldsymbol{w},\boldsymbol{v}} \frac{1}{2}\|\boldsymbol{y} - \boldsymbol{X}\boldsymbol{w} - \boldsymbol{A}\boldsymbol{v}\|^2$$

to construct an estimator. Using Lemma E.3, we have

$$\hat{\boldsymbol{w}}_1 = \boldsymbol{H}_{a\perp}\boldsymbol{y} \quad \text{and} \quad \hat{\boldsymbol{v}}_1 = \boldsymbol{G}_{x\perp}\boldsymbol{y}.$$

Using Lemma E.4, we can simplify

$$\hat{\boldsymbol{w}}_1 = \boldsymbol{H}_{a\perp}\boldsymbol{y} = \boldsymbol{w}^\star + \boldsymbol{0} + \boldsymbol{H}_{a\perp}\varepsilon$$

and

$$\hat{\boldsymbol{v}}_1 = \boldsymbol{G}_{x\perp}\boldsymbol{y} = \boldsymbol{0} + \boldsymbol{v}^\star + \boldsymbol{G}_{x\perp}\varepsilon.$$

Since $\boldsymbol{A}$ is not available at prediction time, we impute it instead with its mean $\mu(\boldsymbol{X})$, which is assumed to be perfectly known. This leads to

$$\tau_{\text{PI}}(\boldsymbol{X}) = \boldsymbol{X}\hat{\boldsymbol{w}}_1 + \mu(\boldsymbol{X})\hat{\boldsymbol{v}}_1 = \boldsymbol{X}\boldsymbol{w}^\star + \mu(\boldsymbol{X})\boldsymbol{v}^\star + \boldsymbol{K}\varepsilon,$$

with

$$\boldsymbol{K} = \boldsymbol{X}\boldsymbol{H}_{a\perp} + \mu(\boldsymbol{X})\boldsymbol{G}_{x\perp}.$$

Plugging into (8) and taking the expectation, we obtain

$$\begin{aligned}
\mathbb{E}[\mathcal{R}(\tau_{\text{PI}}(\boldsymbol{X}))] &= \frac{1}{n}\|\boldsymbol{0}\|^2 + \frac{1}{n}\mathbb{E}[\|\boldsymbol{K}\varepsilon\|^2] \\
&= \frac{\sigma^2}{n}\mathbb{E}[\|\boldsymbol{K}\|^2].
\end{aligned} \tag{10}$$

### E.3 CAPTURING THE BENEFIT OF PI WITH MARGINALIZATION

We then describe when, with marginalization, ordinary least squares ignoring PI is worse than ordinary least squares using PI.

**Proposition E.2.** *Assume that $\boldsymbol{X}^\top\boldsymbol{X}$ is invertible. Moreover, assume that $\boldsymbol{A}^\top\boldsymbol{A}$ and $[\boldsymbol{X},\boldsymbol{A}]^\top[\boldsymbol{X},\boldsymbol{A}]$ are almost surely invertible. We have that*

$$\mathbb{E}[\mathcal{R}(\tau_{\text{marg. NO-PI}}(\boldsymbol{X}))] > \mathbb{E}[\mathcal{R}(\tau_{\text{marg. PI}}(\boldsymbol{X}))]$$

*if and only if*

$$\frac{1}{n}\|(\boldsymbol{I} - \boldsymbol{\Pi}_x)\mu(\boldsymbol{X})\boldsymbol{v}^\star\|^2 + \frac{\sigma^2 d}{n} > \frac{\sigma^2}{n}\|\mathbb{E}[\boldsymbol{L}]\|^2$$

*with $\boldsymbol{L} = \boldsymbol{X}\boldsymbol{H}_{a\perp} + \boldsymbol{A}\boldsymbol{G}_{x\perp}$. When $m = 1$ (i.e., $\boldsymbol{A}$ is a column vector), a sufficient condition is*

$$\frac{1}{n}\|(\boldsymbol{I} - \boldsymbol{\Pi}_x)\mu(\boldsymbol{X})\boldsymbol{v}^\star\|^2 > 2\mathbb{E}\left[\frac{\|\boldsymbol{\Pi}_x\boldsymbol{A}\|^2 + \|\boldsymbol{A}\|^2}{\|(\boldsymbol{I} - \boldsymbol{\Pi}_x)\boldsymbol{A}\|^2}\right] + \frac{\sigma^2 d}{n}.$$

We provide the details of the derivation of the risk for $\tau_{\text{marg. NO-PI}}$ and $\tau_{\text{marg. PI}}$ in Section E.3.1 and Section E.3.2 respectively. Moreover, the second part of the proposition follows from an application of Lemma E.5.

### E.3.1 ORDINARY LEAST SQUARES (WITH MARGINALIZATION)

Restarting from Section E.2.1, we consider the predictions marginalized with respect to $\boldsymbol{A}$. We have

$$\tau_{\text{marg. NO-PI}}(\boldsymbol{X}) = \mathbb{E}_{\boldsymbol{a}\sim p(\boldsymbol{a}|\boldsymbol{x})}[\boldsymbol{X}\hat{\boldsymbol{w}}_0] = \boldsymbol{X}\boldsymbol{w}^\star + \boldsymbol{\Pi}_x\mu(\boldsymbol{X})\boldsymbol{v}^\star + \boldsymbol{\Pi}_x\varepsilon.$$

Plugging into (8), we obtain

$$\mathcal{R}(\tau_{\text{marg. NO-PI}}(\boldsymbol{X})) = \frac{1}{n}\|(\boldsymbol{I} - \boldsymbol{\Pi}_x)\mu(\boldsymbol{X})\boldsymbol{v}^\star - \boldsymbol{\Pi}_x\varepsilon\|^2.$$

Expanding the square and using that $\text{tr}(\boldsymbol{\Pi}_x) = d$, the final risk expression is

$$\begin{aligned}
\mathbb{E}[\mathcal{R}(\tau_{\text{marg. NO-PI}}(\boldsymbol{X}))] &= \frac{1}{n}\|(\boldsymbol{I} - \boldsymbol{\Pi}_x)\mu(\boldsymbol{X})\boldsymbol{v}^\star\|^2 + \frac{1}{n}\mathbb{E}[\|\boldsymbol{\Pi}_x\varepsilon\|^2] \\
&= \frac{1}{n}\|(\boldsymbol{I} - \boldsymbol{\Pi}_x)\mu(\boldsymbol{X})\boldsymbol{v}^\star\|^2 + \frac{\sigma^2 d}{n}.
\end{aligned} \tag{11}$$

### E.3.2 Ordinary least squares with PI and marginalization

Restarting from Section E.2.2, we consider the predictions marginalized with respect to $\boldsymbol{A}$. In particular, we do not impute $\boldsymbol{A}$ by its mean but rather directly take the expectation over $\boldsymbol{A}$. We have

$$\tau_{\text{marg. PI}}(\boldsymbol{X}) = \mathbb{E}_{\boldsymbol{a}\sim p(\boldsymbol{a}|\boldsymbol{x})}[\boldsymbol{X}\hat{\boldsymbol{w}}_1 + \boldsymbol{A}\hat{\boldsymbol{v}}_1] = \boldsymbol{X}\boldsymbol{w}^\star + \mu(\boldsymbol{X})\boldsymbol{v}^\star + \mathbb{E}_{\boldsymbol{a}\sim p(\boldsymbol{a}|\boldsymbol{x})}[\boldsymbol{L}]\varepsilon,$$

with

$$\boldsymbol{L} = \boldsymbol{X}\boldsymbol{H}_{a\perp} + \boldsymbol{A}\boldsymbol{G}_{x\perp}.$$

Plugging into (8) and taking the expectation, we obtain

$$
\begin{aligned}
\mathbb{E}[\mathcal{R}(\tau_{\text{marg. PI}}(\boldsymbol{X}))] &= \frac{1}{n}\|\boldsymbol{0}\|^2 + \frac{1}{n}\mathbb{E}[\|\mathbb{E}_{\boldsymbol{a}\sim p(\boldsymbol{a}|\boldsymbol{x})}[\boldsymbol{L}]\varepsilon\|^2] \\
&= \frac{\sigma^2}{n}\|\mathbb{E}_{\boldsymbol{a}\sim p(\boldsymbol{a}|\boldsymbol{x})}[\boldsymbol{L}]\|^2.
\end{aligned}
\tag{12}
$$

### E.4 Technical lemmas

**Lemma E.3.** *Assume that both $\boldsymbol{X}^\top\boldsymbol{X}$ and $\boldsymbol{A}^\top\boldsymbol{A}$ are invertible. Moreover, assume that both $\boldsymbol{X}_{a\perp}^\top\boldsymbol{X}_{a\perp}$ and $\boldsymbol{A}_{x\perp}^\top\boldsymbol{A}_{x\perp}$ are invertible.*

*We can write the solution of*

$$\min_{\boldsymbol{w},\boldsymbol{v}} \frac{1}{2}\|\boldsymbol{y} - \boldsymbol{X}\boldsymbol{w} - \boldsymbol{A}\boldsymbol{v}\|^2$$

*as*

$$\hat{\boldsymbol{w}} = \boldsymbol{H}_{a\perp}\boldsymbol{y} \quad \text{and} \quad \hat{\boldsymbol{v}} = \boldsymbol{G}_{x\perp}\boldsymbol{y}.$$

*Proof.* The proof follows by applying inversion formula for the block matrix

$$\boldsymbol{Q} = \begin{bmatrix} \boldsymbol{X}^\top\boldsymbol{X} & \boldsymbol{X}^\top\boldsymbol{A} \\ \boldsymbol{A}^\top\boldsymbol{X} & \boldsymbol{A}^\top\boldsymbol{A} \end{bmatrix}$$

where $\boldsymbol{X}_{a\perp}^\top\boldsymbol{X}_{a\perp}$ and $\boldsymbol{A}_{x\perp}^\top\boldsymbol{A}_{x\perp}$ are the two Schur complements of $\boldsymbol{X}^\top\boldsymbol{X}$ and $\boldsymbol{A}^\top\boldsymbol{A}$. Under the assumptions of the lemma, the matrix is $\boldsymbol{Q}$ is invertible. $\qquad\square$

**Lemma E.4.** *We have the following properties*

- $\boldsymbol{H}_{a\perp}\boldsymbol{X} = (\boldsymbol{X}_{a\perp}^\top\boldsymbol{X}_{a\perp})^{-1}\boldsymbol{X}^\top(\boldsymbol{I} - \boldsymbol{\Pi}_a)\boldsymbol{X} = (\boldsymbol{X}_{a\perp}^\top\boldsymbol{X}_{a\perp})^{-1}(\boldsymbol{X}_{a\perp}^\top\boldsymbol{X}_{a\perp}) = \boldsymbol{I}$,

- $\boldsymbol{H}_{a\perp}\boldsymbol{A} = (\boldsymbol{X}_{a\perp}^\top\boldsymbol{X}_{a\perp})^{-1}\boldsymbol{X}^\top(\boldsymbol{I} - \boldsymbol{\Pi}_a)\boldsymbol{A} = \boldsymbol{0}$.

*Conversely, we have*

- $\boldsymbol{G}_{x\perp}\boldsymbol{A} = (\boldsymbol{A}_{x\perp}^\top\boldsymbol{A}_{x\perp})^{-1}\boldsymbol{A}^\top(\boldsymbol{I} - \boldsymbol{\Pi}_x)\boldsymbol{A} = (\boldsymbol{A}_{x\perp}^\top\boldsymbol{A}_{x\perp})^{-1}(\boldsymbol{A}_{x\perp}^\top\boldsymbol{A}_{x\perp}) = \boldsymbol{I}$,

- $\boldsymbol{G}_{x\perp}\boldsymbol{X} = (\boldsymbol{A}_{x\perp}^\top\boldsymbol{A}_{x\perp})^{-1}\boldsymbol{A}^\top(\boldsymbol{I} - \boldsymbol{\Pi}_x)\boldsymbol{X} = \boldsymbol{0}$.

**Lemma E.5.** *Assume $m = 1$, i.e., $\boldsymbol{A}$ is a column vector. We have*

$$\mathbb{E}[\|\boldsymbol{K}\|^2] \le 2d + 2\mathbb{E}\left[\frac{\|\boldsymbol{\Pi}_x\boldsymbol{A}\|^2 + \|\mu(\boldsymbol{X})\|^2}{\|(\boldsymbol{I} - \boldsymbol{\Pi}_x)\boldsymbol{A}\|^2}\right].$$

*Similarly, it holds that*

$$\|\mathbb{E}[\boldsymbol{L}]\|^2 \le 2d + 2\mathbb{E}\left[\frac{\|\boldsymbol{\Pi}_x\boldsymbol{A}\|^2 + \|\boldsymbol{A}\|^2}{\|(\boldsymbol{I} - \boldsymbol{\Pi}_x)\boldsymbol{A}\|^2}\right].$$

*Proof.* We start by splitting the term into

$$\|\boldsymbol{K}\|^2 \le 2\|\boldsymbol{X}\boldsymbol{H}_{a\perp}\|^2 + 2\|\mu(\boldsymbol{X})\boldsymbol{G}_{x\perp}\|^2.$$

Notice that $\boldsymbol{H}_{a\perp}\boldsymbol{H}_{a\perp}^\top = (\boldsymbol{X}_{a\perp}^\top\boldsymbol{X}_{a\perp})^{-1}$ and similarly $\boldsymbol{G}_{x\perp}\boldsymbol{G}_{x\perp}^\top = (\boldsymbol{A}_{x\perp}^\top\boldsymbol{A}_{x\perp})^{-1}$.

Since $\|\boldsymbol{M}\|^2 = \text{tr}(\boldsymbol{M}^\top\boldsymbol{M})$, we have

$$\|\boldsymbol{K}\|^2 \le 2\text{tr}((\boldsymbol{X}^\top\boldsymbol{X})(\boldsymbol{X}_{a\perp}^\top\boldsymbol{X}_{a\perp})^{-1}) + 2\text{tr}(\mu(\boldsymbol{X})^\top\mu(\boldsymbol{X})(\boldsymbol{A}_{x\perp}^\top\boldsymbol{A}_{x\perp})^{-1}).$$

By definition of $\boldsymbol{A}_{x\perp}$, when $m = 1$, we have

$$(\boldsymbol{A}_{x\perp}^{\top}\boldsymbol{A}_{x\perp})^{-1} = \frac{1}{\|(\boldsymbol{I} - \boldsymbol{\Pi}_x)\boldsymbol{A}\|^2}.$$

For the term $(\boldsymbol{X}_{a\perp}^{\top}\boldsymbol{X}_{a\perp})^{-1}$, the Sherman–Morrison formula leads to

$$(\boldsymbol{X}_{a\perp}^{\top}\boldsymbol{X}_{a\perp})^{-1} = (\boldsymbol{X}^{\top}\boldsymbol{X})^{-1} + \frac{1}{1 - \boldsymbol{b}^{\top}(\boldsymbol{X}^{\top}\boldsymbol{X})^{-1}\boldsymbol{b}}(\boldsymbol{X}^{\top}\boldsymbol{X})^{-1}\boldsymbol{b}\boldsymbol{b}^{\top}(\boldsymbol{X}^{\top}\boldsymbol{X})^{-1}$$

with $\boldsymbol{b} = 1/\|\boldsymbol{A}\| \cdot \boldsymbol{X}^{\top}\boldsymbol{A} \in \mathbb{R}^d$. Further simplifying, we obtain

$$\text{tr}((\boldsymbol{X}^{\top}\boldsymbol{X})(\boldsymbol{X}_{a\perp}^{\top}\boldsymbol{X}_{a\perp})^{-1}) = \text{tr}\Big(\boldsymbol{I} + \frac{\boldsymbol{\Pi}_x\boldsymbol{A}\boldsymbol{A}^{\top}\boldsymbol{\Pi}_x}{\|\boldsymbol{A}\|^2 - \|\boldsymbol{\Pi}_x\boldsymbol{A}\|^2}\Big) = d + \frac{\|\boldsymbol{\Pi}_x\boldsymbol{A}\|^2}{\|(\boldsymbol{I} - \boldsymbol{\Pi}_x)\boldsymbol{A}\|^2}.$$

For the second part of the proof, we start by applying Jensen inequality:

$$\|\mathbb{E}[\boldsymbol{L}]\|^2 \le \mathbb{E}[\|\boldsymbol{L}\|^2].$$

The rest of the proof follows along the same arguments, replacing $\mu(\boldsymbol{X})$ by $\boldsymbol{A}$. □

## F    RELATED WORK TABLE

Table 5: Comparison to related work.

| METHOD | $p(\boldsymbol{a}\vert\boldsymbol{x})$ REQUIRED | TRAINING | TEST COST | WEIGHT SHARING | APPROXIMATE $p(y\vert\boldsymbol{x})$ |
|---|---|---|---|---|---|
| IMPUTATION | × | 1 MODEL, 1 STEP | = NO PI | ✓ | × |
| DISTILLATION (LOPEZ-PAZ ET AL., 2015) | × | 2 MODELS, 2 STEPS | = NO PI | × | × |
| HET. DROPOUT (LAMBERT ET AL., 2018) | × | 1 MODEL, 1 STEP | = NO PI | ✓ | ✓ |
| MIML-FCN+ (YANG ET AL., 2017) | × | 1 MODEL, 1 STEP | = NO PI | × | × |
| FULL MARGINALIZATION | ✓ | 1 MODEL, 1 STEP | $\mathcal{O}(S * \text{NO PI})$ | ✓ | ✓ |
| TRAM (OURS) | × | 1 MODEL, 1 STEP | = NO PI | ✓ | ✓ |
| HET-TRAM (OURS) | × | 1 MODEL, 1 STEP | = NO PI | ✓ | ✓ |
| DISTILLED-TRAM (OURS) | × | 2 MODELS, 2 STEPS | = NO PI | ✓ | ✓ |

## G    TWO-STEP TRAM: IMAGENET SCALE REPRESENTATION LEARNING EXPERIMENT

We conduct experiment to test two things: 1) does the one-step TRAM procedure, introduced in §3.2, which is easier for practitioners to implement, approximate the two-step TRAM procedure well and 2) can the results of the toy represrntation learning experiment, §2.2, be replicated in a larger scale setting.

We train a feature extractor with and without access to PI on ImageNet, following the same procedure, architecture and dataset used in the main paper. We then freeze the feature extractor and train a single dense/linear layer with softmax activation on top of the fixed features. We then evaluate the efficacy of these features trained with and without PI using this "linear probe" evaluation widely used in the representation learning literature (Chen et al., 2020).

The results are presented in Table 6. We see that the simpler single-step TRAM method approximates the more complicated two-step TRAM method very well. In addition we see that the features learned by the network with access to PI which are then frozen and evaluated using a linear probe protocol perform better in terms of accuracy and log-likelihood.

## H    TOY EXPERIMENT: VARY $\epsilon$

Table 6: Two-step TRAM: scaling up our toy representation learning experiment. ImageNet validation set negative log-likelihood and accuracy. Averaged over 10 training runs $\pm$ 1 std. dev.

| METHOD | ↓NLL | ↑ACCURACY |
|---|---|---|
| ONE-STEP NO PI | $1.264 \pm 0.007$ | $71.7 \pm 0.2$ |
| TWO-STEP NO PI | $1.265 \pm 0.008$ | $71.7 \pm 0.3$ |
| ONE-STEP TRAM | $1.225 \pm 0.006$ | $72.5 \pm 0.2$ |
| TWO-STEP TRAM | $1.226 \pm 0.002$ | $72.7 \pm 0.2$ |

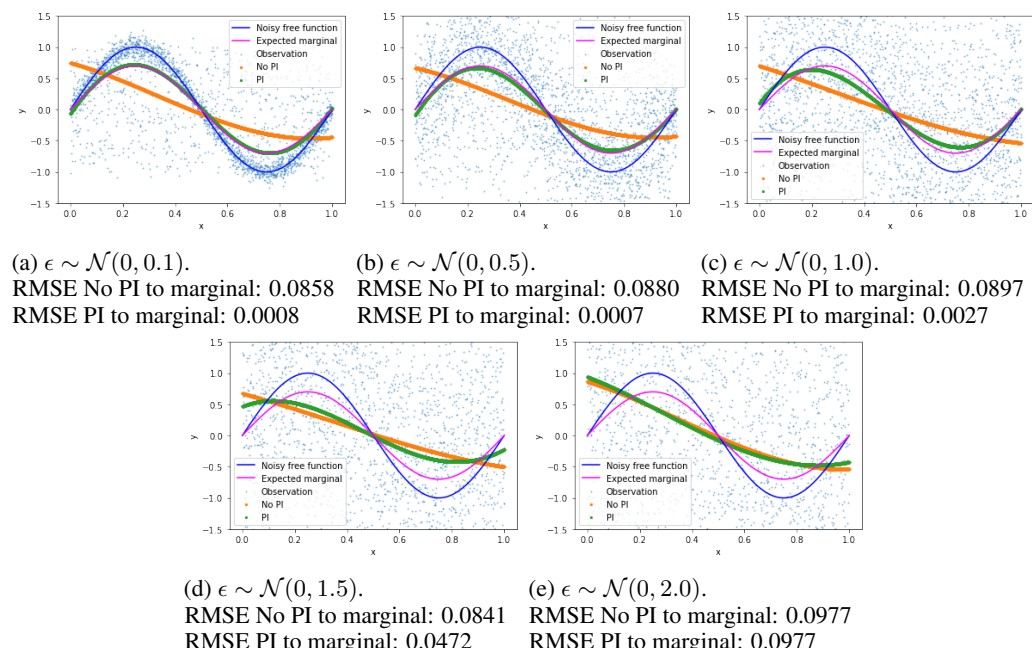

(a) $\epsilon \sim \mathcal{N}(0, 0.1)$.
RMSE No PI to marginal: 0.0858
RMSE PI to marginal: 0.0008

(b) $\epsilon \sim \mathcal{N}(0, 0.5)$.
RMSE No PI to marginal: 0.0880
RMSE PI to marginal: 0.0007

(c) $\epsilon \sim \mathcal{N}(0, 1.0)$.
RMSE No PI to marginal: 0.0897
RMSE PI to marginal: 0.0027

(d) $\epsilon \sim \mathcal{N}(0, 1.5)$.
RMSE No PI to marginal: 0.0841
RMSE PI to marginal: 0.0472

(e) $\epsilon \sim \mathcal{N}(0, 2.0)$.
RMSE No PI to marginal: 0.0977
RMSE PI to marginal: 0.0977

Figure 4: Varying the influence of $\epsilon$ on our motivating toy experiment.

We vary the standard deviation of $\epsilon$ used in our motivating toy experiment, §2.2. The results can be seen graphically in Fig. 4. Fig. 4 also contains the average RMSE to the true marginal across the data points plotted. The graphical and numerical results demonstrate that even for large levels of noise PI aids with representation learning but as expected, as the level of noise grows the advantage of using PI diminishes as it becomes increasingly difficult to distinguish irreducible noise from noise which can be explained away with PI.

# I  IMAGENET EXPERIMENT PI ABLATION

We run an ablation, removing PI feature: the probability of the label assigned by the model from the PI set. We are thus left with just one PI feature, the one-hot encoded ID of the model that produced the label.

We see the results in Table 7. As expected (and predicted by our theoretical analysis), removing informative PI reduces the effectiveness of TRAM. Nonetheless, TRAM with the reduced PI feature set still outperforms the No PI baseline, with accuracy and log-likelihood lying between the No PI and full PI feature set TRAM methods.

Table 7: ImageNet ablation with reduced PI feature set. ImageNet validation set negative log-likelihood and accuracy. Averaged over 10 training runs $\pm$ 1 std. dev.

| METHOD | ↓NLL | ↑ACCURACY |
|---|---|---|
| NO PI | $1.264 \pm 0.007$ | $71.7 \pm 0.2$ |
| TRAM WITH FULL PI SET | $1.225 \pm 0.006$ | $72.5 \pm 0.2$ |
| TRAM WITH REDUCED PI SET | $1.246 \pm 0.004$ | $72.0 \pm 0.2$ |

