# OpenReview forum: "Transfer and Marginalize: Explaining Away Label Noise with Privileged Information"
_ICLR.cc/2022/Conference — ICLR 2022 Submitted_

### Official Review · Reviewer_GTi4 · 2021-10-31

**Correctness:** 3
**Technical Novelty And Significance:** 3
**Empirical Novelty And Significance:** 3
**Recommendation:** 6
**Confidence:** 4

**Main Review:**

Strengths:

1. The proposed framework uses privileged information in deep neural networks efficiently.

2. The method is well-demonstrated with both theoretical analyses and numerical experiments. Thus the result is convincing.

3. Using privileged to deal with label noise may help to better characterize the instance-dependent label noise. Thus this work could contribute a new idea to learning with label noise.


Weaknesses:

1. The notations could be better defined. For example, in Page 2, in addition to saying "Denoting by $\prod_x$ the orthogonal projector associated with $X$", it would be better to show the equation as well. In Page 3, $u$ in $q(y|x,a;u)$ is not clearly defined.It is also confusing why $a \cdot u$ is the noise term. Should it be $a \cdot v$?

2. The privileged information in the toy example may be too strong since $\epsilon$ is too small compared with $v$ and knowing $a$ means knowing which point is corrupted by label noise. The privileged information is often side information to help better decisions, rather than critical knowledge as shown in this example.

3. The feature extractor trained with privileged information is supposed to be better than the one trained without privileged information. But the current experiment did not explicitly show it. An ablation study that compares the performance of fine-tuning the last layer using clean data with fixed feature extractors (trained with PI or not) would be helpful.

**Summary Of The Paper:**

This paper develops a simple and efficient method for training with privileged information and testing without privileged information. The training with privileged information focuses on transferring via sharing the weights of the encoder, which is the knowledge learned with privileged information.
The testing without privileged information addresses the practical case where test data may not be equipped with useful privileged information. The corresponding solution is to marginalize over privileged information at test time.

**Summary Of The Review:**

The paper proposes a new perspective on the noisy learning literature. The proposed framework is simple but useful.

---

> ### Author Response · Authors · 2021-11-17
> **Response to Reviewer GTi4**
>
> We thank the reviewer for their time, helpful comments and suggestions. We have added several new experiments and made significant changes to the paper to address your comments. All new additions to the paper are highlighted in orange - many are included in the appendix.
>
> **Re: the reviewer’s first point on improved notation.**
>
> We thank the reviewer for pointing out these areas of improvement. We have now referenced the definition of the orthogonal projector in Page 2. We have updated the definition of q(y|x, a, u) in Page 3. And yes the reviewer is correct that the a.u noise term was a typo, it has now been fixed, see Page 3.
>
> **Re: The privileged information in the toy example may be too strong since eps is too small compared with v**
>
> Thanks for this excellent suggestion. We have added an ablation varying the value of eps in Appendix H (Figure 4). The graphical and numerical results demonstrate that even for large levels of noise, PI aids with representation learning but as expected, as the level of noise (eps) grows, the advantage of using PI diminishes as it becomes increasingly difficult to distinguish irreducible noise from noise which can be explained away with PI.
>
> **Re: An ablation study that compares the performance of fine-tuning the last layer using clean data with fixed feature extractors**
>
> Thanks again for this really nice suggestion. We have added precisely such an experiment in Appendix G, where we scale up our toy representation learning experiment to ImageNet scale. We show that representations learned with PI, which are then frozen and evaluated using a linear classification model on ImageNet, outperform representations learned without access to PI. (Table 6, Appendix G).

---

> > ### Comment · Reviewer_GTi4 · 2021-11-25
> > **Thanks for the response. One quick question**
> >
> > Thanks for the responses. Most of my concerns are addressed.
> >
> > **One quick question:** Did the author test the generalization accuracy of training with the clean CIFAR-10 test dataset? I find the test accuracy on 50k training images could be 83% when the model is trained on the clean test dataset. The noise rate of the data in Table 1 seems to be only 20%. It is not very intuitive to observe a 16% drop (83%-->67%) with only 20% noise rates.

---

> > > ### Author Response · Authors · 2021-11-25
> > > **Training on clean CIFAR-10 test set**
> > >
> > > We thank the reviewer for taking the time to review our response.
> > >
> > > We have run the additional requested experiment. When we train on the CIFAR-10 (clean) test set and evaluate on the CIFAR-10 (clean) training set we get an accuracy of 73.0% +- 0.80, averaged over 10 random seeds.
> > >
> > > We have used exactly the same hyperparameters as in the paper i.e. pre-trained MobileNet, training for 20 epochs with the Adam optimizer, etc. The gap in the accuracy between training on the clean test set and the noisy CIFAR-10H dataset agrees with the reviewer's intuition about the approximate effect of the noise rate from CIFAR-10H.

---

> > > > ### Author Response · Authors · 2021-11-30
> > > > **Summary**
> > > >
> > > > We thank the reviewer for their positive feedback and we hope they will update their score accordingly. If there are any further concerns we can address please let us know.

---

> > > > > ### Comment · Reviewer_GTi4 · 2021-12-01
> > > > > **thanks for the rebuttals**
> > > > >
> > > > > Thank you for the detailed rebuttal. I will keep my initial evaluation (6).

---

### Official Review · Reviewer_KQ8f · 2021-11-01

**Correctness:** 2
**Technical Novelty And Significance:** 3
**Empirical Novelty And Significance:** 2
**Recommendation:** 3
**Confidence:** 3

**Main Review:**

Strengths:
1. Propose the notion of using PI to improve the performance of a network by explaining away the noise from annotators.
2. Propose a new way to implement that idea using a stop gradient operator

Weaknesses:
1. Theoretically, the authors do not give a formal proof of why using PI in a practical network like ResNet helps to improve performance. The authors just give the proof for the very simple edge cases like linear and non-linear sine models. How about the classification case that this paper is targeted for? These simple cases are only true when the feature extraction network is kept unchanged during training.

2. Empirically, the experiments are conducted in a limited and counter-intuitive: (a) When testing on CIFAR-10, it is quite unintuitive that the authors train on the test set and evaluate on the training set. To address the lack of training examples since the test set is used for training, they use the pretrained model on ImageNet; (b) When testing on the synthetic dataset relabeled from ImageNet, the information of the probability of the label assigned by a pretrained model is used as a proxy for the confidence of a human annotator. This information tells a lot about the quality of the label, which is unrealistic in real-world scenarios since, for each example, the annotators are required to additionally answer the confidence question resulting in a significant cost and ambiguity.

My suggestion is that the authors should collect a real-world large-scale dataset such that the PI is insignificantly expensive to obtain along with the main annotations to prove their claim empirically. Otherwise, this paper has a limited impact on practical scenarios.

**Summary Of The Paper:**

This paper suggests a notion that using additional information called privilege information (PI) from annotators will help explain away the noise of the label they annotated and propose a way to implement that idea. The authors give the intuition from a simple linear model and a non-linear sine model that PI is useful. Then, an implementation of that idea called TRAM is proposed to make use of the information from PI through a stop gradient operator in training.  Since there are scant large-scale datasets with PI annotation, to evaluate the proposed approach, the authors have to create a synthetic one relabeled from ImageNet.

**Summary Of The Review:**

This paper proposes an intuitive phenomenon that using PI is helpful and improves the performance of the model. However, the authors have not given any proof for the general case (not the edge cases) theoretically or empirically successfully.

---

> ### Author Response · Authors · 2021-11-17
> **Response to Reviewer KQ8f**
>
> We thank the reviewer for their time, helpful comments and suggestions. We have added several new experiments and made significant changes to the paper to address your comments. All new additions to the paper are highlighted in orange - many are included in the appendix.
>
> **Re: our use of the probability of the label assigned by a pretrained model**
>
> We thank the reviewer for this comment. The intention for using this feature was to simulate other features such as; the experience of a human annotator, the time taken by an annotator to provide a label, etc. which would be available at no additional cost in realistic settings (i.e. would be available without asking the annotator for them directly).
>
> Nonetheless, we have added an additional ablation to the paper in which we remove the label probability feature from the ImageNet PI feature set. (Table 7, Appendix G). As expected removing this feature reduces the performance of the TRAM method, but given just the model ID feature, TRAM still outperforms the No PI baseline.
>
> **Re: When testing on CIFAR-10, it is quite unintuitive that the authors train on the test set and evaluate on the training set.**
>
> CIFAR-10H only has labels with annotator features for the CIFAR-10 test set, therefore to leverage PI at training time it is necessary to train on the official CIFAR-10 test set and evaluate on the training set, we have no other option. Please note however that there are no examples which are used for both training and testing so the correct experimental protocol is followed.
>
> In addition, we note that we have added a qualitative analysis of the CIFAR-10H results, Sec. 5.2, main paper. This analysis shows that the TRAM method uses PI to effectively downweight the importance of unreliable human annotators. This reduces the harmful effect of noisy labels, which is then transferred via weight sharing to the output head without access to PI. (Figure 3, Sec. 5.2, main paper).
>
> **Re: the authors do not give a formal proof of why using PI in a practical network like ResNet helps to improve performance**
>
> We agree with the reviewer that a formal proof for a ResNet style model would be ideal, however there are two key challenges in the analysis of real deep-learning models for classification:
>
> (a) The non-convexity/absence of simple characterisation of the solutions
>
> (b) The loss: the risk analysis in the classification case is still a research topic, even for linear models (e.g.,. Francis Bach. "Self-concordant analysis for logistic regression." Electron. J. Statist. 4 384 - 414, 2010)
>
> Given these constraints, our goal is to motivate the usage of PI in a setup where the analysis is tractable and some take-home messages can be nonetheless conveyed to the real settings.

---

> > ### Comment · Reviewer_KQ8f · 2021-11-19
> > **Thanks for your response**
> >
> > 1. The improvement in accuracy in Tab 7 of TRAM with the reduced PI Set is insignificant compared to that of TRAM with the Full PI set (0.3% vs. 0.8%). And of course, this might be due to the additional parameters needed to learn the $\psi$. Also, for a very large dataset like ImageNet, simulating it with only 16 models is not realistic. How about simulating with 50, 100, or more models? For each model in the 16 models, you can get some variants of it by choosing checkpoints of different epochs or changing some hyper-parameters. It would be interesting to have more annotators as an additional ablation for this TRAM. Intuitively, more and more annotator IDs will diminish the effect of the denoising model using this information.
> >
> > 2. What I want to say here is the usage of the pretrained ImageNet. 10,000 labeled images are sufficient to train a small and efficient network like MobileNet with an image size of 32 x 32. It would be better to disentangle the effect of the pretraining model on the denoising method like this. Do you have any ablations without pretraining MobileNet on ImageNet to compare these results?
> >
> > 3. So you confirm that these proofs are just for motivation purposes, not a theoretical contribution?

---

> > > ### Author Response · Authors · 2021-11-22
> > > **Thank you for your response**
> > >
> > > We thank the reviewer for taking the time to review our response and to provide good suggestions to further strengthen the experiments.
> > >
> > > 1. Yes, we agree that if you remove the label probability feature the accuracy on ImageNet is lower than with this feature. This is to be expected given the less informative PI feature set. We note however that the test time network which is being evaluated **has the exact same number of parameters as the No PI baseline**.
> > > Further we thank the reviewer for their suggestion of simulating the ImageNet relabelling with more than 16 models. We note that in the main paper the ImageNet PI feature set has the real-valued label probability feature, which is different for each (sample, annotator) pair; so in fact, the PI feature set domain is already infinite.
> > > To better study the hypothesis suggested by the reviewer (namely “Intuitively, more and more annotator IDs will diminish the effect of the denoising model using this information”), we would like to highlight that there are two factors at play: (i) the total number of annotators and (ii) the number of annotations per sample. We believe it is interesting to both look at how the performance varies with respect to (i), keeping (ii) fixed, and conversely (in real-world applications though, varying (ii) is generally costly). We can simulate more annotators by following the scheme suggested by the reviewer (e.g., different checkpoints, different hyperparameters).
> > > In addition, note that the CIFAR-10H experiment uses 82 annotators.
> > >
> > > 2. We will follow-up with an ablation removing pre-training from the CIFAR-10H experiment. Note that the ImageNet and CivilComments experiments do not use pre-trained models.
> > >
> > > 3. We reiterate that the formal proofs are a theoretical contribution, in that they are a novel analysis, not available in the literature, describing the effect of PI within a linear regression model. We use this theoretical contribution to motivate our method, better understand TRAM and when/why PI helps improve the test time performance, despite being only available at training time. As with any theory, it is necessary to make assumptions which may not always be fulfilled in practice. We believe that extending our current theoretical contribution to the setting of deep-learning classifiers is an exciting, though challenging, avenue for future work.

---

> > > > ### Comment · Reviewer_KQ8f · 2021-12-05
> > > > **Thanks to the authors for their responses**
> > > >
> > > > The proof of the simple linear and cosine cases is not aimed at the main purpose of the paper which is about the deep network such as ResNet. So in my opinion, I don't consider it as a contribution.
> > > > Also, the paper also lacks some important ablation study which I listed above also have a negative impact on the faithfulness of the paper when applying to the practical scenarios. I have waited for their results for about two weeks and no results were reported.
> > > > Therefore, I decided to lower my score to three.

---

### Official Review · Reviewer_6G97 · 2021-11-02

**Correctness:** 3
**Technical Novelty And Significance:** 3
**Empirical Novelty And Significance:** 2
**Recommendation:** 6
**Confidence:** 3

**Main Review:**

Strengths
1. The proposed methods have simple architectures (not requiring specific modules, e.g., Gaussian dropout [Lambert et al., 2018], for the marginalization).
2.  Making prediction for a test point (PI is not available) has the same cost with a standard network trained without PI.

Weaknesses
The manuscript could be improved in presentation. I am listing some points which are unclear and need to be addressed.
- What is the formal definition of label noise for classification? It seems that the current version of the manuscript provides an example for label noise for regression (Section 2.2), but there is not given a clear definition or an example for label noise for classification.
- I think that the transition from the theoretical analysis using the linear regression model in Section 2 to the motivation of approximating the conditional marginal distribution is not that smooth. It seems that the analysis just supports that PI can be useful. In fact, the motivation of approximating the conditional marginal distribution is directly explained by the equation (2) (the idea of the representation learning with PI is explained by the example in 2.2). I think that the section 2.1 could be revised to address how this theoretical analysis can motivate the approximation of the conditional marginal distribution.
- In the introduction, it is stated “We provide empirical evidence suggesting that the representations learned with access to PI are more robust against label noise”. What does “empirical evidence” refer to? The motivating example in Section 2.2?  If the answer is yes, I am not sure if this toy example can be convincing empirical evidence. I think that this is just a motivating example to explain the intuitive idea of the representational learning with PI. In addition, the example is for regression (the assumptions here would not be realistic settings for classification).


**Summary Of The Paper:**

The manuscript proposes classification methods utilizing privileged information (PI, which is available only at training time). The main idea is to directly approximate the conditional marginal distribution, i.e., p(y|x) \approx q(y|\pi(x), w), where \pi(x) is a feature extractor and w is the weight parameter. Another main point is to learn the representation (\pi(x)) of the features with access to PI at the training time. To do this, the proposed methods employ knowledge transfer by weight sharing. The proposed methods include several variants by adopting heteroscedastic classification (Het-TRAM) or distillation (Distilled-TRAM).

**Summary Of The Review:**

I think that the main idea, directly approximating the conditional marginal distribution without requiring any sampling methods or additional specific architectures, is interesting (although I think the proposed methods are not super novel). I also like the way the authors build their methods from theoretical analyses and motivating examples. However, I at the same time think that the manuscript could be improved. Please see the detailed comments above.

---

> ### Author Response · Authors · 2021-11-17
> **Response to Reviewer 6G97**
>
> We thank the reviewer for their time, helpful comments and suggestions. We have added several new experiments and made significant changes to the paper to address your comments. All new additions to the paper are highlighted in orange - many are included in the appendix.
>
> **Re: the formal definition of label noise for classification.**
>
> We agree with the reviewer that label noise takes on multiple definitions in the literature. For completeness, we have added two references to Sec 1 of the main paper of recent surveys of the label noise literature which discuss those definitions. [A, B]. For our work, we are interested in label noise in the sense defined by Lemma 1.1 in Sec 1, i.e. we are interested in residual entropy in the labels Y which is unexplained by the standard features X but can be partially explained away by the privileged information A. Formally, H(Y | X) - H(Y | X, A).
>
> **Re: section 2.1 could be revised to address how this theoretical analysis can motivate the approximation of the conditional marginal distribution.**
>
> We thank the reviewer for this suggestion. We highlight that the marginalization case is discussed fully in Appendix E and is signposted at the end of Sec. 2.1. Due to space constraints, unfortunately we could not move the entire discussion from the appendix to the main paper. If the reviewer believes that more space should be dedicated to this part of the analysis, we can of course more deeply refactor the camera-ready version of the paper to integrate this content.
>
> **Re: empirical evidence suggesting that the representations learned with access to PI are more robust against label noise**
>
> Thank you for challenging us on this. We have added a large-scale (ImageNet) representation learning experiment to the paper (Appendix G, Table 6). In this new experiment we scale up our toy representation learning experiment to ImageNet scale. We follow the same procedure as for our toy representation learning experiment but for ImageNet with a ResNet50. As requested this is also a classification task, so we have extended our toy representation learning experiment in terms of scale (2D -> ImageNet) and task (regression -> classification).
>
> We show that representations learned with PI, which are then frozen and evaluated using a linear classification model on ImageNet, outperform representations learned without access to PI.
>
> In addition, we add a qualitative analysis of the CIFAR-10H results, Sec. 5.2, main paper. This analysis probes why TRAM helps with representation learning by being more robust to label noise.  As demonstrated, TRAM uses PI to effectively downweight the importance of unreliable human annotators. This reduces the harmful effect of noisy labels, which is then transferred via weight sharing to the output head without access to PI. (Figure 3, Sec. 5.2, main paper).
>
> [A] Song, Hwanjun, et al. "Learning from noisy labels with deep neural networks: A survey." arXiv preprint arXiv:2007.08199 (2020).
>
> [B] Cordeiro, Filipe R., and Gustavo Carneiro. "A Survey on Deep Learning with Noisy Labels: How to train your model when you cannot trust on the annotations?." 2020 33rd SIBGRAPI Conference on Graphics, Patterns and Images (SIBGRAPI). IEEE, 2020.

---

> > ### Comment · Reviewer_6G97 · 2021-12-02
> > **Thanks to the authors for their responses**
> >
> > Most of my questions are addressed. I read the other reviewers' comments and the authors' responses. I also decided to stick with my initial evaluation (6: marginally above the acceptance threshold).

---

### Official Review · Reviewer_Nesa · 2021-11-02

**Correctness:** 3
**Technical Novelty And Significance:** 2
**Empirical Novelty And Significance:** 2
**Recommendation:** 6
**Confidence:** 3

**Main Review:**

Strength:
1. the authors present a clear explanation of the underlying intuition of using PI and the motivation behind TRAM.
2. the idea of integrating privileged information through weight sharing is interesting, and the two-step design is novel (as far as I am concerned).
3. the proposed method can (in principle) apply to any neural network model and has zero overhead at prediction time.

Weakness/Concerns:

1. In the title and abstract, the authors state that privileged information helps explain away label noise. However, the latter investigations only focus on the accuracy improvement without an in-depth discussion about how and how much the label noise can be "explained away." in realistic deep neural network models.
2. there lacks quantitative (or even qualitative) evidence about how, how much, and what kind of privileged information is transferred through weight sharing in realistic deep neural network models.
3. In the discussion of Table 2, it is overstated that "TRAM provides significant performance improvement" since a ~5% increase in accuracy does not seem significant at all.

Continuing point 2 the above, I have a question: Is it possible to control the amount of privileged information transferred to the shared network block by designing a specific training scheme? Say (just a rough idea), we may gradually increase the weight of the shared network block during backpropagate through the $q(y|x,a)$ branch.

**Summary Of The Paper:**

This paper proposes a method, TRAM, that integrates the privileged information into the learned network weight through weight sharing at training time and approximately marginalizes over the privileged information at test time. Experiments on both realistic and synthetic datasets demonstrate the effectiveness of TRAM in improving model performance on noisy data.

**Summary Of The Review:**

The paper proposes TRAM, a method of integrating privileged information in the network weights during training. Experiments on both realistic and synthetic datasets show that TRAM can help improve the model accuracy on noisy data. However, the improvement is significant. Further, the lack of detailed investigations about how the privileged information is integrated into the network weights and how the privileged information help explain away the label noise in realistic deep neural network models limits the contribution of the present work.

---

> ### Author Response · Authors · 2021-11-17
> **Response to Reviewer Nesa**
>
> We thank the reviewer for their time, helpful comments and suggestions. We have added several new experiments and made significant changes to the paper to address their comments. All new additions to the paper are highlighted in orange - many are included in the appendix.
>
> **Re: quantitative (or even qualitative) evidence about how, how much, and what kind of privileged information is transferred through weight sharing in realistic deep neural network models.**
>
> We have added three new experiments to address this concern and better understand both quantitatively and qualitatively how and how much PI is transferred in realistic deep networks:
>
> 1) In Appendix G we scale up our toy representation learning experiment to ImageNet scale. We show that representations learned with PI, which are then frozen and evaluated using a linear classification model on ImageNet, outperform representations learned without access to PI. (Table 6, Appendix G).
> 2) In Appendix I, we provide an ablation, removing the label probability features from the ImageNet PI feature set. (Table 7, Appendix G). The final results and conclusions are not altered by the removal of this feature.
> 3) We add a qualitative analysis of the CIFAR-10H results, Sec. 5.2, main paper. This analysis shows that the TRAM method uses PI to effectively downweight the importance of unreliable human annotators. This reduces the harmful effect of noisy labels, which is then transferred via weight sharing to the output head without access to PI. (Figure 3, Sec. 5.2, main paper).
>
> Further we thank the reviewer for their comment regarding our use of the term “significant” in discussing the CIFAR-10H results. We have toned down our statement and the corresponding Sec. 5.1 has been updated accordingly.
>
> **Re: Is it possible to control the amount of privileged information transferred to the shared network block by designing a specific training scheme? Say (just a rough idea), we may gradually increase the weight of the shared network block during backpropagate through the q(y|x,a) branch.**
>
> We thank the reviewer for their interesting idea, we would be happy for the reviewer to expand on their comment. If we understand correctly, does the reviewer mean that the stop_gradient in the TRAM method could be replaced with a weighted gate with value [0, 1]? If so, certainly that could be done, but it is not clear to us what benefit this would bring.
>
> We further note that the choice of the architecture for the channel that takes PI as input, i.e., Psi(phi(x), .), can be designed to be more or less expressive and therefore capture a different "amount of privileged information".

---

> > ### Comment · Reviewer_Nesa · 2021-12-05
> > **Thanks for the responses**
> >
> > Most of my concerns are addressed. After reading the revised paper and extended appendix, I decided to increase my score to 6: marginally above the acceptance threshold.

---

### Author Response · Authors · 2021-11-17
**Response Summary**

We thank the reviewers for their detailed and helpful comments. Several comments have motivated additional experiments and other improvements to the text. We will respond in detail to each reviewer individually and provide a short global summary of the main additions here.

We have added 4 new experiments in response to reviewer comments, these are ablations and analyses of the pre-existing experiments:

1) In Appendix G we scale up our toy representation learning experiment to ImageNet scale. We show that representations with access to PI, which are then frozen and evaluated using a linear classification model on ImageNet, outperform representations learned without access to PI. (Table 6, Appendix G).
2) In Appendix I, we provide an ablation, removing the label probability features from the ImageNet PI feature set. (Table 7, Appendix G). The final results and conclusions are not altered by the removal of this feature.
3) We add a qualitative analysis of the CIFAR-10H results, Sec. 5.2, main paper. This analysis shows that the TRAM method uses PI to effectively downweight the importance of unreliable human annotators. This reduces the harmful effect of noisy labels, which is then transferred via weight sharing to the output head without access to PI. (Figure 3, Sec. 5.2, main paper).
4) We have added an ablation varying the value of eps in Appendix H (Figure 4). The graphical and numerical results demonstrate that even for large levels of noise, PI aids with representation learning but as expected, as the level of noise (eps) grows, the advantage of using PI diminishes as it becomes increasingly difficult to distinguish irreducible noise from noise which can be explained away with PI.

We have also fixed some typos and presentation issues the reviewers pointed out and responded to any questions reviewers had individually.

All new additions to the paper are highlighted in orange - most are included in the appendix.

---

### Decision · Program_Chairs · 2022-01-20

**Decision:**

Reject

**Comment:**

The manuscript proposes (TRansfer and Marginalize) TRAM method that integrates the privileged information into the learned network weight through weight sharing at training time and approximately marginalizes over the privileged information at test time. TRAM can also be combined with methods for dealing with noisy labels, distillation (Distilled-TRAM) and heteroscedastic output layers (Het-TRAM). Experiments are performed on both realistic and synthetic datasets including CIFAR-10H, re-labeled ImageNet, and Civil Comments Identities.

Reviewers agreed on several positive aspects of the manuscript, including:
1. The proposed methods have simple architectures (not requiring specific modules, e.g., Gaussian dropout [Lambert et al., 2018], for the marginalization);
2. The proposed method can in principle be applied to any neural network model and has zero overhead at prediction time.

Reviewers also highlighted several major concerns, including:
1. The analysis is performed on edge cases such as linear and non-linear sine models. There is no analysis for the classification case that this manuscript is targeted for. The simple cases are only true when the feature extraction network is kept unchanged during training;
2. Empirically, the experiments are conducted in a limited and counter-intuitive;
3. Lack empirical evidence suggesting that the representations learned with access to privileged information are more robust against label noise;
4. Lack quantitative (or even qualitative) evidence about how, how much, and what kind of privileged information is transferred through weight sharing in realistic deep neural network models.

Several new experiments have been added to show, among others: representations learned with privileged information outperform representations learned without access to privileged information (using a linear classification model on ImageNet), better quantitatively and qualitatively understanding how and how much privileged information is transferred in realistic deep networks.

Post-rebuttal, reviewers stayed with borderline ratings, and they have suggested further improvements: simulating with more annotators by using different checkpoints and/or different hyperparameters, collecting a real-world large-scale dataset such that the privileged information is insignificantly expensive to obtain along with the main annotations, and disentangling the effect of the pretraining model on the denoising method.